# SSNA1 stabilizes dynamic microtubules and detects microtubule damage

**Elizabeth J Lawrence[1], Goker Arpag[1], Cayetana Arnaiz[1], Marija Zanic[1,2,3]***

[1]Department of Cell and Developmental Biology, Vanderbilt University, Nashville, United States; [2]Department of Chemical and Biomolecular Engineering, Vanderbilt University, Nashville, United States; [3]Department of Biochemistry, Vanderbilt University, Nashville, United States

**Abstract** Sjögren's syndrome nuclear autoantigen-1 (SSNA1/NA14) is a microtubule-associated protein with important functions in cilia, dividing cells, and developing neurons. However, the direct effects of SSNA1 on microtubules are not known. We employed in vitro reconstitution with purified proteins and TIRF microscopy to investigate the activity of human SSNA1 on dynamic microtubule ends and lattices. Our results show that SSNA1 modulates all parameters of microtubule dynamic instability—slowing down the rates of growth, shrinkage, and catastrophe, and promoting rescue. We find that SSNA1 forms stretches along growing microtubule ends and binds cooperatively to the microtubule lattice. Furthermore, SSNA1 is enriched on microtubule damage sites, occurring both naturally, as well as induced by the microtubule severing enzyme spastin. Finally, SSNA1 binding protects microtubules against spastin's severing activity. Taken together, our results demonstrate that SSNA1 is both a potent microtubule-stabilizing protein and a novel sensor of microtubule damage; activities that likely underlie SSNA1's functions on microtubule structures in cells.

## Editor's evaluation

In this manuscript, Lawrence et al. investigate the direct effects of the microtubule-associated protein, Sjögren's Syndrome Nuclear Autoantigen 1 (SSNA1), on microtubule dynamics and damage using purified proteins and TIRF microscopy. The authors show that SSNA1 is a microtubule stabilizing protein that acts to slow rates of growth and shrinkage, and promote rescue. Furthermore, SSNA1 serves as a sensor of microtubule damage and protects microtubules from the microtubule severing enzyme, spastin. This paper will be of broad interest to scientists interested in cytoskeletal cell biology.

*For correspondence:
marija.zanic@vanderbilt.edu

**Competing interest:** The authors declare that no competing interests exist.

## Introduction

Sjogren's syndrome nuclear autoantigen-1 (SSNA1/NA14) is a microtubule-associated protein (MAP) that plays important roles in cilia, cell division, and neuronal development. In cilia, SSNA1 localizes to basal bodies and axonemes where it is required for proper cilium assembly and intraflagellar transport (*Lai et al., 2011*; *Pfannenschmid et al., 2003*; *Schoppmeier et al., 2005*). In dividing cells, SSNA1 is enriched at the spindle poles and midbody, and is necessary for proper cell division (*Goyal et al., 2014*; *Pfannenschmid et al., 2003*). Finally, SSNA1 promotes axon elongation and branching in developing neurons (*Basnet et al., 2018*; *Goyal et al., 2014*). Although SSNA1 is involved in a range of microtubule-driven cellular processes, its direct effects on microtubules are not known.

SSNA1 is a small (~14 kDa), coiled-coil protein that self-assembles into higher-order fibrils (*Basnet et al., 2018*; *Ramos-Morales et al., 1998*; *Rodríguez-Rodríguez et al., 2011*). A recent in vitro study using cryo-EM/ET reported that SSNA1 fibrils bind longitudinally along stabilized microtubules,

induce microtubule branching, and promote microtubule nucleation (*Basnet et al., 2018*). Growing microtubules undergo dynamic instability; a phenomenon whereby individual microtubules alternate between phases of growth and shrinkage via the transitions referred to as catastrophe and rescue (*Mitchison and Kirschner, 1984*). Given its direct interaction with microtubules and localization to sites of dynamic microtubule growth in cells, SSNA1 is well-positioned to impact microtubule dynamics. Nonetheless, whether SSNA1 regulates dynamic microtubules has not been investigated.

Microtubule regulation is not restricted to the dynamic microtubule ends. For example, microtubule-severing enzymes, motor proteins, and mechanical forces induce microtubule lattice damage (*Théry and Blanchoin, 2021*). The damaged microtubule lattice can be recognized and stabilized by MAPs (*Aher et al., 2020*; *Aumeier et al., 2016*; *de Forges et al., 2016*; *Schaedel et al., 2015*; *Schaedel et al., 2019*; *Théry and Blanchoin, 2021*; *Triclin et al., 2018*; *Vemu et al., 2018*). SSNA1 has been identified as a binding partner of spastin, a microtubule-severing enzyme (*Errico et al., 2004*); and both SSNA1 and spastin localize to spindle poles and neuronal branch points (*Basnet et al., 2018*; *Goyal et al., 2014*; *Yu et al., 2008*). However, whether SSNA1 regulates microtubule lattice damage remains an open question.

In this study, we employed in vitro reconstitution techniques with purified protein components and TIRF microscopy to interrogate the roles of SSNA1 in regulating dynamic microtubule ends and microtubule lattice damage.

## Results

### Human SSNA1 suppresses microtubule dynamicity

To investigate the effects of SSNA1 on dynamic microtubules, we purified human SSNA1 protein (*Figure 1—figure supplement 1*) and employed an established TIRF-based microtubule dynamics in vitro reconstitution assay (*Gell et al., 2010*). Previous work implicated SSNA1 in spontaneous microtubule nucleation (*Basnet et al., 2018*). To build upon these observations, we assessed the ability of SSNA1 to promote templated microtubule nucleation from GMPCPP-stabilized microtubule seeds, which better reflects microtubule nucleation in cells (*Wieczorek et al., 2015*; *Figure 1A*). We performed a titration of soluble tubulin from 3 μM to 10 μM with and without 2.5 μM SSNA1 and found that SSNA1 promoted templated nucleation compared to the tubulin alone condition, thus supporting the role of SSNA1 in microtubule nucleation (*Figure 1B*).

To assess the direct effects of SSNA1 on microtubule dynamics, we performed SSNA1 titration experiments in which microtubules were grown with a range of SSNA1 concentrations (*Figure 1C*, *Video 1*). While the average cellular concentration of SSNA1 is estimated to be ~200 nM (HeLa cells; *Itzhak et al., 2016*), given that the SSNA1 localization is highly restricted to centrosomes, basal bodies, and axonal branch points (*Basnet et al., 2018*; *Goyal et al., 2014*; *Pfannenschmid et al., 2003*), the effective local concentration of SSNA1 is likely significantly higher. Therefore, we investigated the effects of up to 3 μM SSNA1 on microtubule dynamics (*Figure 1—figure supplement 2*). We found that SSNA1 suppressed microtubule dynamicity (defined as the total length of growth and shrinkage divided by the total time spent in growth and shrinkage) at both plus and minus ends (*Figure 1D*). Further, by quantifying the individual microtubule dynamic parameters, we found that at the lowest concentrations tested, SSNA1 primarily suppressed microtubule catastrophe and shrinkage rates (*Figure 1E and F*), while at higher concentrations SSNA1 additionally suppressed microtubule growth rate and promoted microtubule rescue (*Figure 1—figure supplement 2*). Therefore, we demonstrate that SSNA1 is a microtubule-stabilizing protein that suppresses microtubule dynamicity by modulating all parameters of dynamic instability.

### The progressive slowdown in microtubule growth correlates with SSNA1 accumulation on microtubule ends

Interestingly, we observed that the microtubule growth rate appeared to slow down over time when microtubules were grown in the presence of SSNA1. To investigate this further, we tracked the ends of microtubules grown in the presence of 3 μM SSNA1 for up to 30 min and calculated the velocity of the microtubule ends over time (*Figure 2—figure supplement 1*). Quantification of the microtubule end velocity revealed a statistically significant slowdown in the mean microtubule velocity (from 7.6 ± 0.9 nm/s at 3 min to 3.1 ± 0.8 nm/s at 15 min, mean ± SEM, N=23, p<0.001, unpaired t-test).

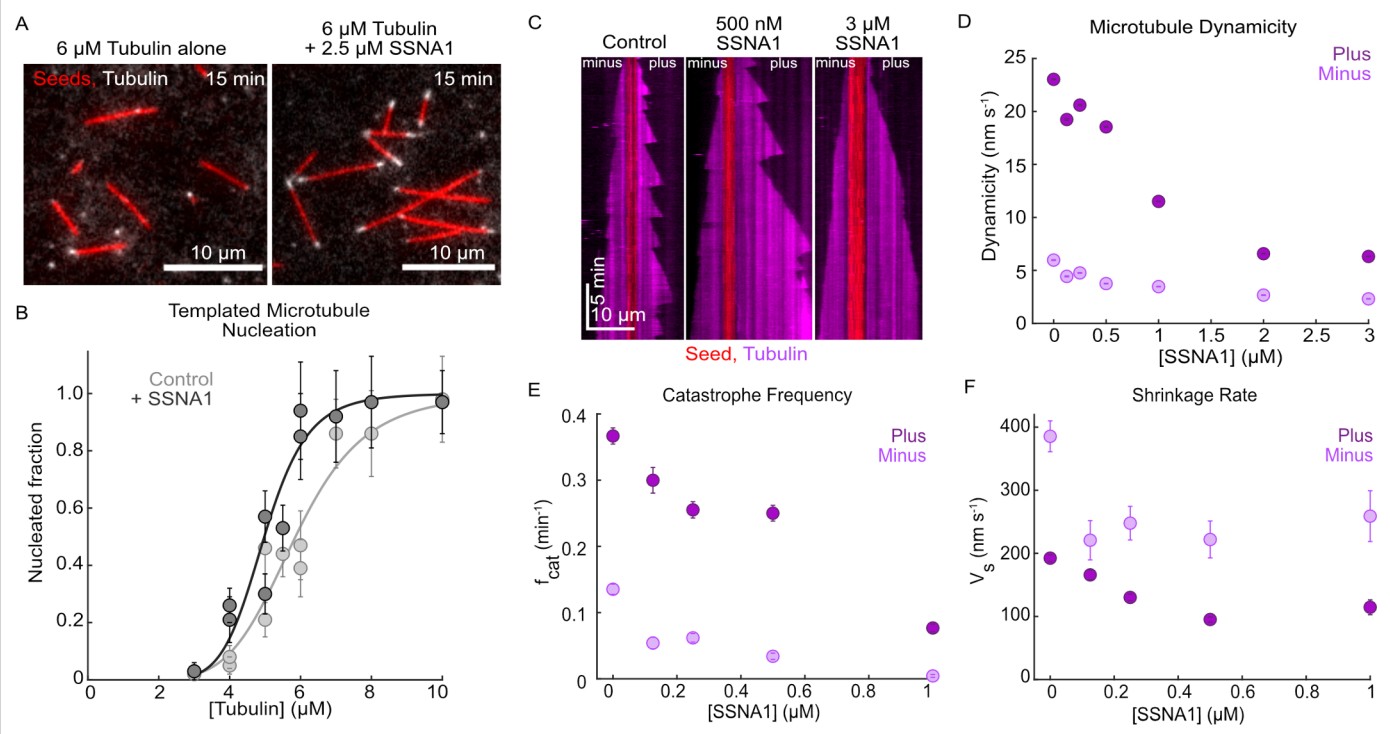

**Figure 1.** Human SSNA1 promotes microtubule nucleation and suppresses microtubule dynamicity. (**A**) Representative images of a templated microtubule nucleation assay in which microtubule extensions (gray) were nucleated from GMPCPP-stabilized seeds (red) in the presence and absence of SSNA1. Images shown are for the 6 μM tubulin condition with and without 2.5 μM 488-SSNA1 at 15 min after the introduction of the nucleation reaction. (**B**) Quantification of the fraction of seeds that nucleated in 15 min with tubulin alone (control, light gray) and 2.5 μM 488-SSNA1 (dark gray) as a function of tubulin concentration. Data are individual experimental replicates ±SE from six experimental days (N=30–68 microtubules for each concentration tested in the tubulin-alone control condition, N=33–77 microtubules for each concentration tested in the SSNA1 condition). The data were fitted to a sigmoidal curve of the form y(x)=xs/(C+xs) (solid lines). For tubulin alone, C=5.9 μM (95% CI: 5.5–6.2) and s=6.0 (95% CI: 3.4–8.6). For the SSNA1 condition, C=5.0 μM (95% CI: 4.7–5.3) and s=8.0 (95% CI: 4.3–11.9). (**C**) Representative kymographs of microtubules grown from GMPCPP-stabilized seeds with 9 μM Alexa-647 tubulin alone (control) and in the presence of 500 nM and 3 μM SSNA1. The microtubule plus ends are shown on the right and the minus ends are shown on the left. (**D**) Quantification of microtubule dynamicity as a function of the SSNA1 concentration. Dynamicity is calculated as the total length of growth and shrinkage over the observation time. (**E**) Quantification of the microtubule catastrophe frequency at the plus and minus ends of microtubules grown with 9 μM tubulin and concentrations of SSNA1 from 0 μM to 1 μM. (**F**) Quantification of the microtubule shrinkage rate at the plus and minus ends of microtubules grown with 9 μM tubulin and concentrations of SSNA1 from 0 μM to 1 μM. For the quantifications in panels (**D–F**), data are weighted means ± SE obtained from four independent experimental days (N=43–485 growth events for each concentration tested at microtubule plus ends; N=30–120 growth events for each concentration tested at microtubule minus ends). Plus end data are in dark purple; minus end data are in light purple.

The online version of this article includes the following source data and figure supplement(s) for figure 1:

**Source data 1.** An Excel sheet containing numerical data for the quantification of microtubule nucleation and microtubule dynamics presented in Figure 1 and Figure 1 - figure supplement 2.

**Figure supplement 1.** Human SSNA1 protein purification.

**Figure supplement 2.** SSNA1 modulates all parameters of microtubule dynamics.

To visualize SSNA1 localization on growing microtubule ends we chemically labeled purified SSNA1. First, we confirmed that labeling did not interfere with SSNA1's ability to self-assemble into fibrils (*Figure 2—figure supplement 2*). Next, we used our TIRF microscopy assay and observed SSNA1 localization on growing microtubules over time (*Figure 2A*, *Video 2*). We tracked the ends of microtubules grown in the presence of 5 μM labeled SSNA1 (*Figure 2B*) and calculated the velocity of the microtubule ends over time (*Figure 2C*). Once again, we found that the majority of microtubule ends experienced a significant slowdown in growth velocity (from 8.4 ± 1.4 nm/s at 3 min to 5.0 ± 0.7 nm/s at 15 min, mean ± SEM, N=16 and N=25, respectively, p=0.01, unpaired t-test), although individual microtubules displayed a notable variability in both the timing and the rate of growth slowdown. Quantitative measurements of SSNA1 intensity at the microtubule end region revealed that SSNA1

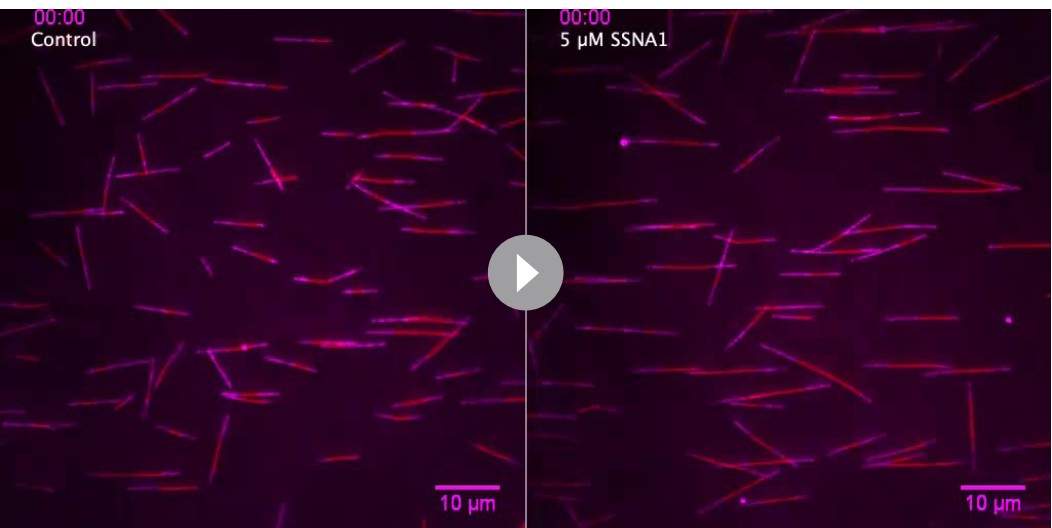

**Video 1.** SSNA1 stabilizes dynamic microtubules. Dynamic microtubules were grown from GMPCPP-stabilized seeds with 9 µM Alexa-647 tubulin alone (control, left) and in the presence of 5 µM SSNA1 (right). The seeds are shown in red and the dynamic extensions are in magenta. Time is min:s. Scale, 10 µm. Playback, 30 fps.
https://elifesciences.org/articles/67282/figures#video1

intensity increased over time (*Figure 2D*) (from 0.33 ± 0.07 a.u. at 3 min to 0.67 ± 0.09 a.u. at 15 min, mean ± SEM, N=16 and N=25, respectively, p=0.01, unpaired t-test) and inversely correlated with the microtubule end velocity (*Figure 2E*). However, on an individual microtubule level, we observed that the slowdown in growth rate often coincided with local SSNA1 enrichment at the microtubule tip (*Figure 2A*, yellow arrow). Therefore, the microtubule growth rate progressively slows down in the presence of SSNA1, but the extent and onset of the slowdown vary between individual microtubules.

## SSNA1 forms stretches along growing microtubule ends

Our observations of SSNA1 accumulation on the ends of growing microtubules raised the question of whether SSNA1 recognizes the nucleotide state of tubulin, preferentially binding to the GTP-tubulin cap at growing microtubule ends. To investigate whether SSNA1 has a binding preference for specific microtubule lattice regions, we performed wash-in experiments in which we first grew dynamic microtubule extensions with 15 µM tubulin from GMPCPP-stabilized seeds in the absence of SSNA1 and then introduced 15 µM tubulin and 2.5 µM 488-SSNA1 into the reaction mix ( *Figure 3A, B*, *Video 3*). This experimental setup allowed us to determine whether SSNA1 preferred to bind to GMPCPP-stabilized microtubule seeds (thought to mimic the GTP-cap); pre-existing, old microtubule lattices (GDP); or new microtubule lattices that were grown in the presence of SSNA1. Measurements of the mean SSNA1 intensity revealed that the SSNA1 binding was similar on GMPCPP-seeds versus the pre-existing GDP lattice, indicating that SSNA1 does not specifically recognize the tubulin nucleotide state (*Figure 3C*). In contrast, we observed enhanced stretches of SSNA1 intensity that occurred predominantly on the new lattice and initiated at growing microtubule ends (*Figure 3B, D*).

Over time, the majority of growing microtubule ends exhibited a pronounced stretch of SSNA1 (81%, or 59 out of 73 microtubules analyzed from three independent experiments), at either one or both microtubule ends (*Figure 3E*). Interestingly, we observed that the SSNA1 stretches expanded over time, and these expansions proceeded exclusively in the direction of microtubule growth at both microtubule ends (*Figure 3F*). In addition, we found that SSNA1 stretches could resolve, allowing the microtubule end to continue to grow dynamically (*Figure 3—figure supplement 1*). SSNA1 stretches could subsequently serve as stable rescue sites on the microtubule lattice (*Figure 3—figure supplement 1*). Taken together, we conclude that SSNA1 may recognize a specific structural feature that appears at growing microtubule ends, and that such structural feature can persist along the newly polymerized microtubule lattice, resulting in stretches of SSNA1 accumulation.

Additionally, we observed that ends of microtubules grown in the presence of SSNA1 could become highly curled over time (*Figure 3G, H*, *Videos 1 and 2*). Although SSNA1 was enriched on curled

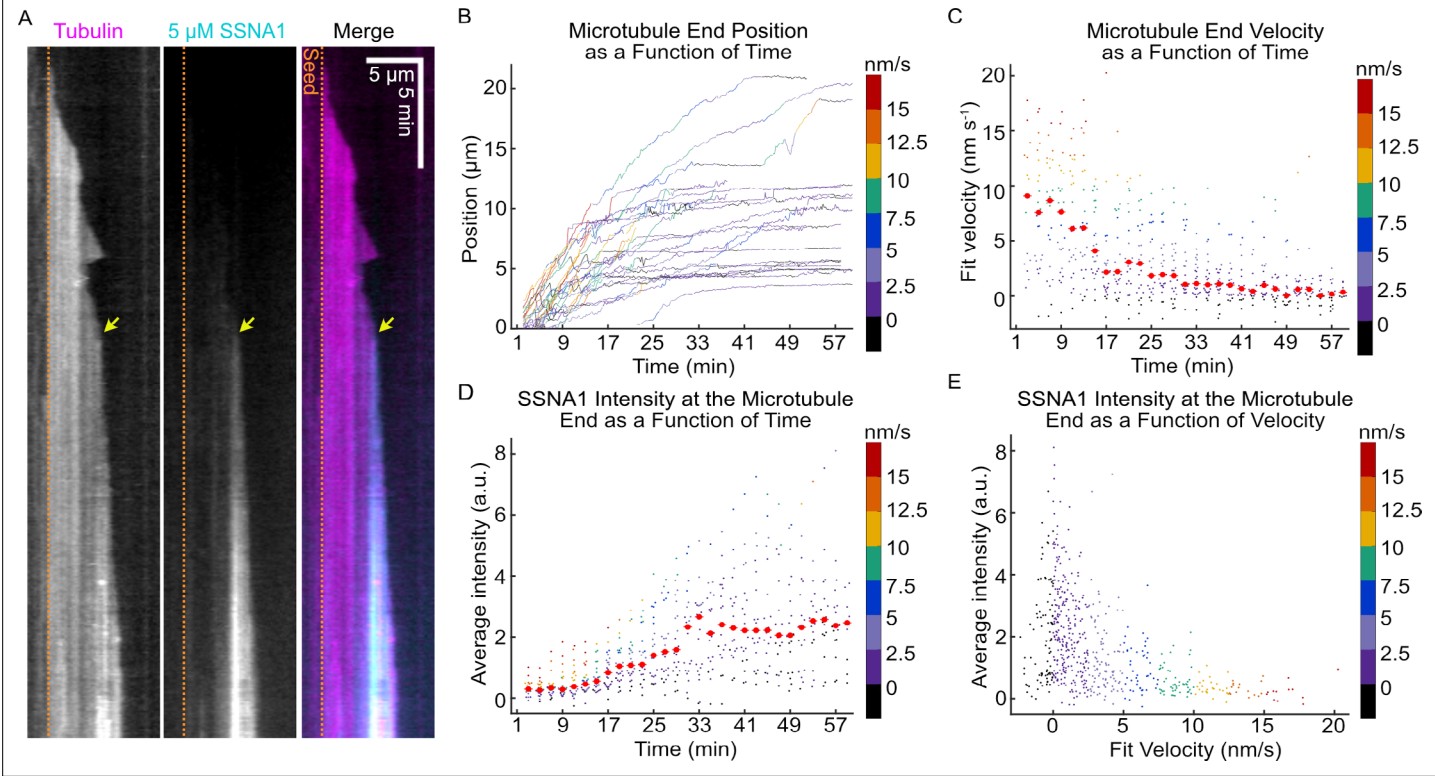

**Figure 2.** The progressive slowdown in microtubule growth correlates with SSNA1 accumulation on microtubule ends. (**A**) A representative kymograph of a microtubule grown with 9 µM Alexa-647-labeled tubulin and 5 µM Alexa-488-labeled SSNA1 (7% labeled) showing the progressive localization of 488-SSNA1 to the microtubule end region over time. The yellow arrow indicates SSNA1 enrichment at the microtubule end at the onset of growth slowdown. The dotted orange lines demarcate the position of the microtubule seed (not shown). (**B**) Microtubule end positions as a function of time for microtubules grown with 9 µM tubulin and 5 µM Alexa-488-labeled SSNA1, color-coded with 2-min segment velocity. (**C**) Microtubule end velocities over a 2-min segment as a function of time calculated from the end positions in (**B**) and color-coded with velocity for consistency. Plots of microtubule end velocities and corresponding SSNA1 fluorescence intensities at microtubule ends (**D**) as a function of time and (**E**) as a function of 2-min segment velocity. A total of 26 microtubules were analyzed. For the plots in (**D**) and (**E**), the median for each bin is shown as a bright red point with horizontal line.

The online version of this article includes the following source data and figure supplement(s) for figure 2:

**Source data 1.** An Excel sheet containing numerical data for the quantification of microtubule end position and velocity over time, and SSNA1 intensity as a function of both time and microtubule end velocity, as presented in Figure 2 and Figure 2 - figure supplement 1.

**Figure supplement 1.** Microtubule growth rate slows down over time in the presence of SSNA1.

**Figure supplement 2.** Labeled SSNA1 forms fibrils.

ends, our quantitative analysis revealed that the SSNA1 intensity did not correlate with the degree of microtubule curvature (*Figure 3I*). To further investigate whether SSNA1 recognizes regions of increased microtubule curvature, we assessed its localization on Taxol-stabilized microtubules, which inherently display a broad range of curvatures when polymerized with tubulin alone in the absence of SSNA1 (*Figure 3J, K*). We found that SSNA1 does not specifically localize to curved microtubule regions (*Figure 3L*). We thus conclude that SSNA1 is not a sensor of microtubule curvature, but rather induces microtubule curling at growing microtubule ends.

## SSNA1 binds cooperatively to microtubules

To further probe the kinetics of SSNA1-microtubule binding, we investigated SSNA1 localization on stabilized microtubules. The binding of 1 µM labeled SSNA1 to GMPCPP-stabilized microtubules showed that the SSNA1 fluorescence intensity increased linearly over time across all microtubule lattices in the field of view until the SSNA1 signal saturated at ~5 min (*Figure 4—video 1*, *Figure 4— figure supplement 1*). Using a low concentration of labeled SSNA1 (5 nM) allowed us to probe microtubule binding on a single-molecule level. Quantification of the single-molecule dwell times of labeled SSNA1 on GMPCPP-stabilized microtubules revealed that individual SSNA1 molecules bound to the

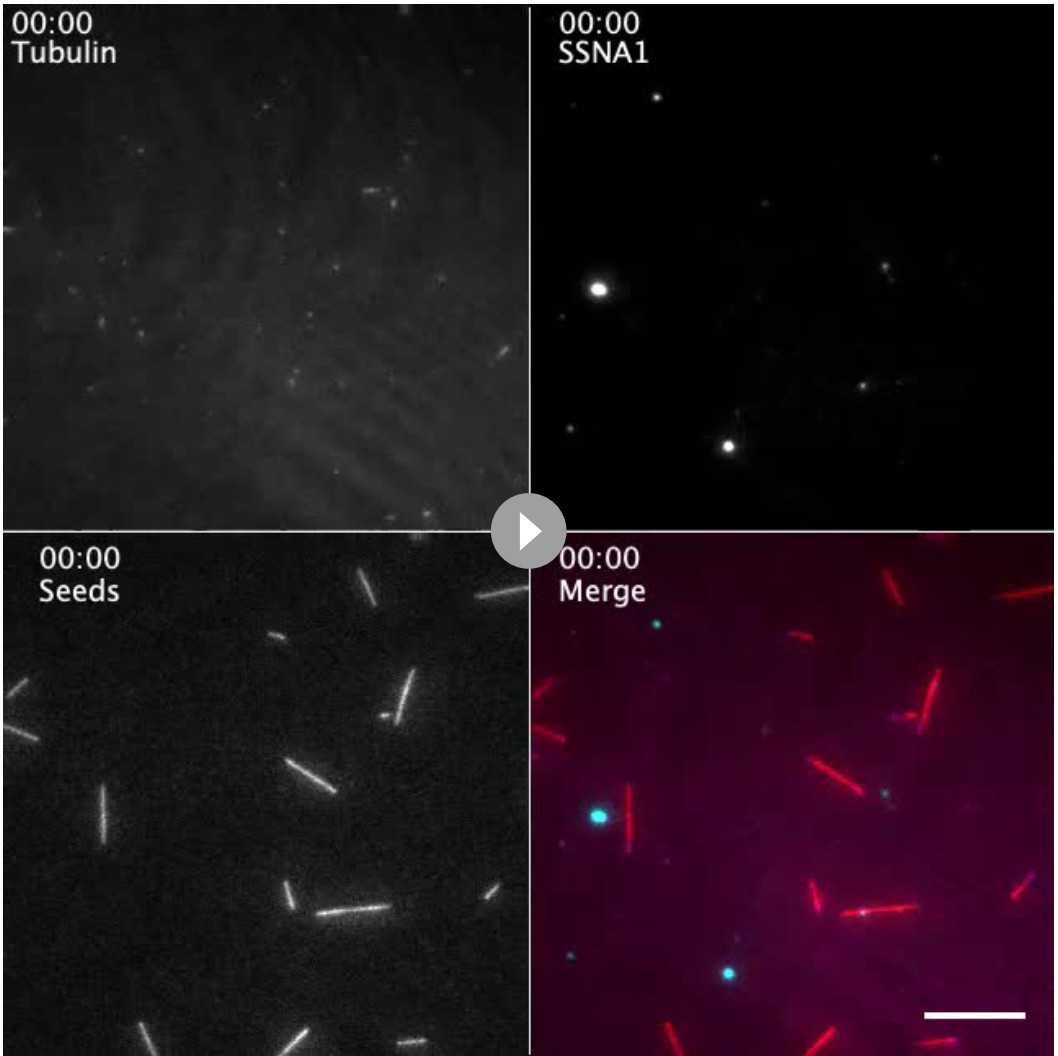

**Video 2.** SSNA1 progressively localizes to dynamic microtubule extensions. Dynamic microtubule extensions (magenta) were grown from GMPCPP-stabilized microtubule seeds (red) with 9 µM tubulin and 5 µM 488-SSNA1 (cyan). Time is in min:s. Scale, 10 µm. Playback, 30 fps.
https://elifesciences.org/articles/67282/figures#video2

microtubule with mean durations of 12 ± 2 s (SE, N=142) during the first 5 min after SSNA1 was introduced. The observed dwell times of single molecules remained the same in the consecutive 5 min after SSNA1 introduction (10 ± 2 s, SE, N=107, p=0.2 Wilcoxon rank test) (*Figure 4A, B*). To investigate potential cooperativity between SSNA1 molecules in a higher concentration regime, we performed single-molecule 'spiking' experiments, probing the 647-SSNA1 single-molecule dwell times in the background of 1 µM 488-SSNA1 (*Figure 4C*). Interestingly, we found that the single-molecule dwell times were longer in the presence of excess SSNA1 with a mean dwell time of 19 ± 3 s (SE, N=144) in the first 5 min, which further increased to 39 ± 4 s (SE, N=294, p<0.001 compared to the single-molecule control, Wilcoxon rank test) in the second 5 min after SSNA1 introduction (*Figure 4C–E*). Additionally, quantification of SSNA1 binding events revealed that SSNA1 association rates in the spiking conditions also increased over time from $(0.96 \pm 0.08) \times 10^{-3}$ events $\mu m^{-1} nM^{-1} s^{-1}$ to $(2.4 \pm 0.1) \times 10^{-3}$ events $\mu m^{-1} nM^{-1} s^{-1}$ (p<0.001, Welch's t-test). Combined, the differences in binding kinetics between the single-molecule control and spiking conditions demonstrate cooperativity in SSNA1 localization on the microtubule lattice. These results suggest that SSNA1 molecules assemble over time into higher-order structures on the microtubule lattice, consistent with previous reports (*Basnet et al., 2018*; *Rodríguez-Rodríguez et al., 2011*).

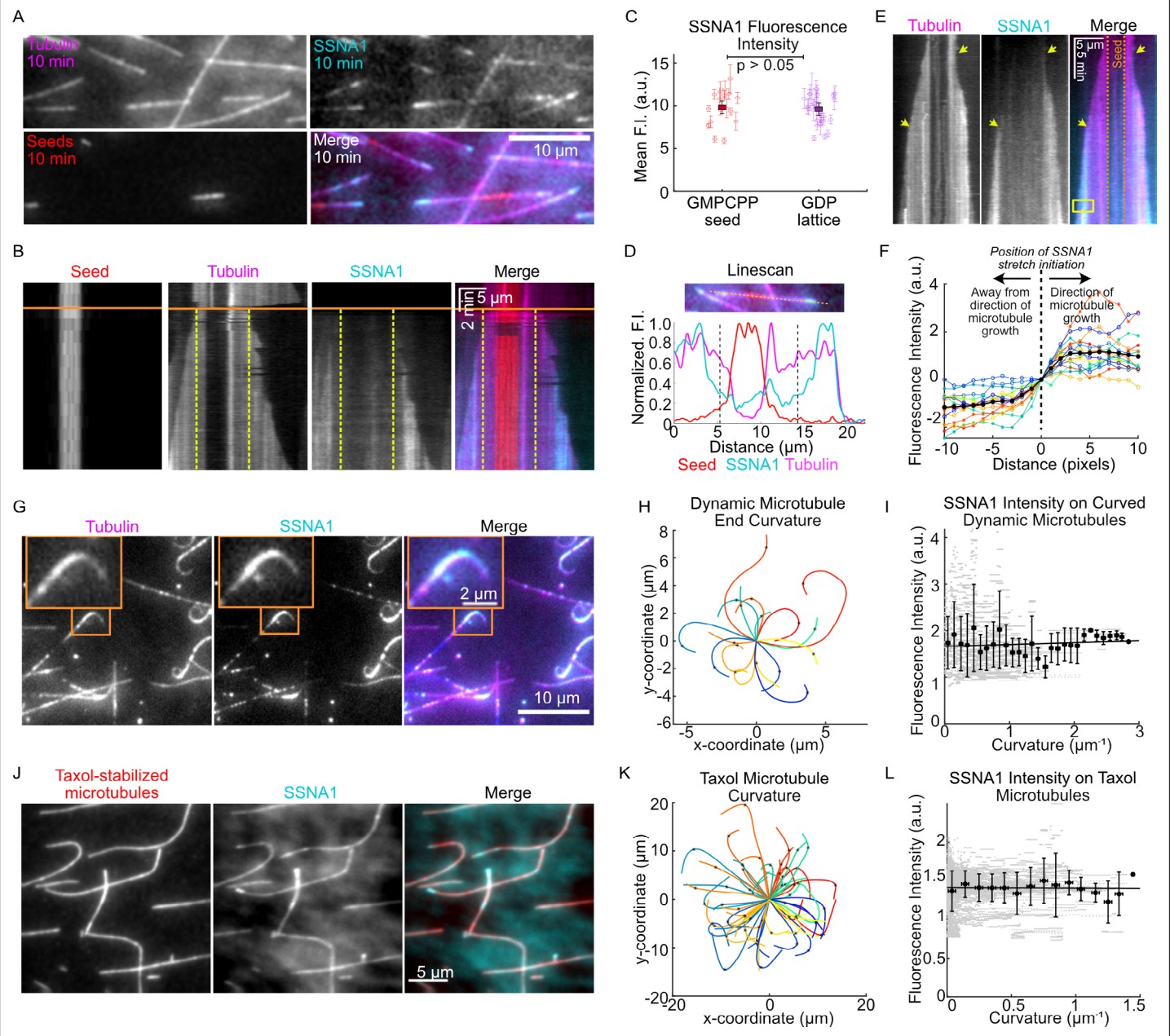

**Figure 3.** SSNA1 forms stretches on growing microtubule ends. (**A**) A representative field of view of microtubules at 10 min post-SSNA1 wash-in. Microtubule extensions were pre-grown with 15 µM 647-tubulin and then 15 µM tubulin and 2.5 µM 488-SSNA1 (15% labeled) were introduced into the channel. (**B**) A kymograph of a microtubule from a wash-in experiment. The solid orange line indicates the time of SSNA1 introduction. The dashed vertical lines mark the boundary between the pre-existing lattice and the new lattice. (**C**) Quantification of the mean SSNA1 fluorescence intensity on the GMPCPP-stabilized microtubule seeds and pre-existing GDP microtubule lattices. A total of 17 microtubules were analyzed. Statistical significance was determined by Welch's t-test. (**D**) Linescans showing the normalized fluorescence intensities of the microtubule seed (red), dynamic microtubule extension (magenta), and SSNA1 (cyan). The two dashed vertical lines mark the boundary between the pre-existing and new microtubule lattice, as indicated on the kymograph in (**B**). (**E**) An example kymograph showing stretches of SSNA1 forming at both microtubule ends. (**F**) Quantification of the SSNA1 fluorescence intensity toward and away from the direction of microtubule growth. The vertical dotted line indicates the position on the lattice at which the SSNA1 stretch initiated. N=16 SSNA1 stretches were analyzed. (**G**) Representative images of curved microtubules that were grown with 9 µM tubulin and 5 µM 488-labeled SSNA1 (7% labeled) for 60 min. The orange box indicates the zoomed region shown in inset. (**H**) Plots of curved microtubule extensions of microtubules grown in the conditions described for (**G**). A total of 17 curls were analyzed. (**I**) Corresponding quantification of SSNA1 intensity as a function of microtubule curvature on dynamic microtubules. Individual data points are in gray and the means and SD of binned data (0.1 µm⁻¹ bin width) are in black. The solid line is a linear fit to the means and the slope is not significantly different from zero (slope= 0.04 [95% CI: –0.05 to 0.13] a.u. × µm, p-value=0.4). (**J**) Taxol-stabilized microtubules were incubated with 5 µM 488-SSNA1 and imaged for 60 min. Sum

*Figure 3 continued on next page*

*Figure 3 continued*

projection images of a representative field of view are shown. (**K**) Plots of curved Taxol-stabilized microtubules. A total of 59 microtubules were analyzed. (**L**) Corresponding quantification of SSNA1 intensity as a function of microtubule curvature on Taxol-stabilized microtubules. Individual data points are in gray and the means and SD of binned data (0.1 μm$^{-1}$ bin width) are in black. The solid line is a linear fit to the means and the slope is not significantly different from zero (slope = −0.006 [95% CI: −0.118 to 0.105] a.u. × μm, p-value=0.9).

The online version of this article includes the following source data and figure supplement(s) for figure 3:

**Source data 1.** An Excel sheet containing numerical data for the quantifications presented in Figure 3.

**Figure supplement 1.** SSNA1 stretches can resolve and serve as stable rescue sites.

## SSNA1 detects microtubule damage

Our observations of SSNA1 stretches on dynamic microtubules suggested that SSNA1 may detect a specific structural feature of the growing microtubule end. Interestingly, we noticed that SSNA1 intensity was enhanced in regions of lower tubulin intensity on both Taxol-stabilized and GMPCPP-stabilized microtubules (*Figure 5—figure supplement 1*). We also observed SSNA1 enrichment in regions of lower tubulin intensity on dynamically growing microtubules (*Figure 3—figure supplement 1*). Because regions of lower tubulin intensity may represent sites of microtubule lattice damage, we hypothesized that SSNA1 recognizes microtubule damage. To test this hypothesis, we asked whether SSNA1 recognizes microtubule damage induced by spastin, a microtubule severing enzyme that plays important roles in regulating the dynamics and organization of microtubule networks (*Kuo and Howard, 2021*; *McNally and Roll-Mecak, 2018*). We induced microtubule damage by pre-incubating stabilized microtubules with 100 nM purified human spastin (*Figure 5—figure supplement 2*) for 5 min and then exchanged the reaction for a solution containing 5 μM labeled SSNA1. We observed that SSNA1 was specifically enriched on regions of the microtubule lattice that had lower tubulin fluorescence intensity (*Video 4*, *Figure 5A*). Kymograph and linescan analysis of the SSNA1 and tubulin fluorescence intensities demonstrated that SSNA1 progressively accumulated at the sites of microtubule damage over the course of the experiment (*Figure 5*). Therefore, we conclude that SSNA1 is a novel sensor of both naturally occurring and spastin-induced microtubule lattice damage.

## SSNA1 protects microtubules against spastin-mediated microtubule severing

Given that SSNA1 stabilizes growing microtubule ends and recognizes microtubule lattice damage, we wondered whether SSNA1 can protect the microtubule lattice against the severing activity of spastin. To address this, we incubated GMPCPP-stabilized microtubules with or without 1 μM 488-SSNA1 for 10 min and then introduced 200 nM spastin. We found that, while spastin was able to efficiently sever control microtubules, no severing was observed for the SSNA1-coated microtubules over 30 min (*Figure 6—figure supplement 1*, *Video 5*). To further elucidate the interplay of spastin with SSNA1, we used a GFP-labeled truncated (del227) human spastin construct, which retains full severing

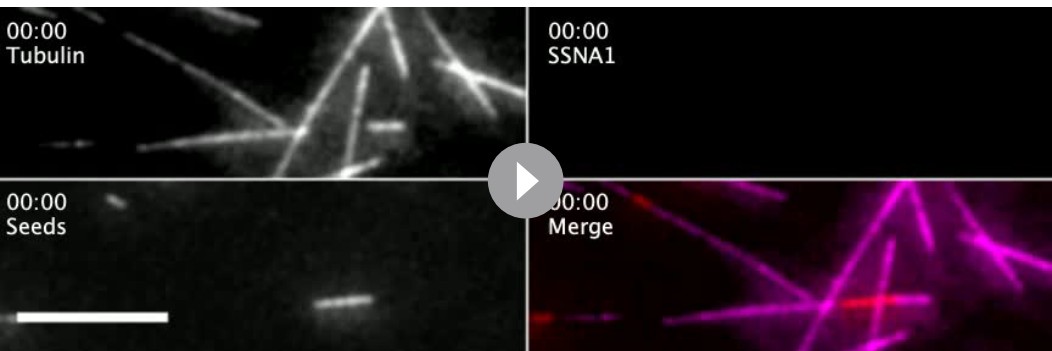

**Video 3.** SSNA1 forms stretches on growing microtubule ends. A field of view of microtubules in a wash-in experiment in which dynamic microtubule extensions (magenta) were initially grown from GMPCPP-stabilized microtubule seeds (red) with 15 μM tubulin alone and subsequently, at t=2 min, the reaction was exchanged to 15 μM tubulin and 2.5 μM 488-SSNA1 (cyan). Time is in min:s. Scale, 10 μm. Playback, 30 fps.
https://elifesciences.org/articles/67282/figures#video3

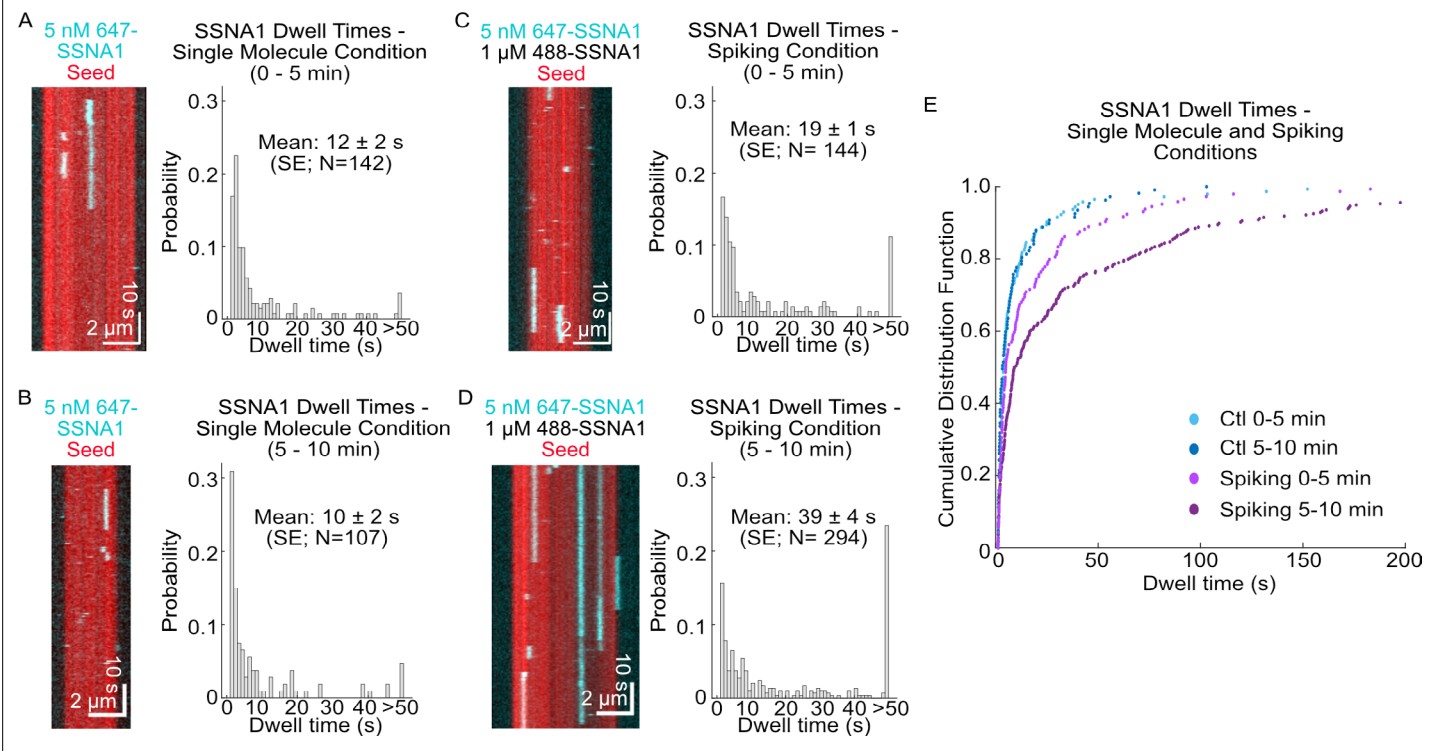

**Figure 4.** SSNA1 binds cooperatively to microtubules. Representative kymographs of single-molecule SSNA1 binding events on GMPCPP-stabilized microtubules and corresponding quantification of the single-molecule dwell times (**A**) 0–5 min and (**B**) 5–10 min after SSNA1 addition. Representative kymographs of single-molecule SSNA1 binding events in the presence of excess SSNA1 ('spiking' condition) on GMPCPP-stabilized microtubules and corresponding quantification of the single-molecule dwell times (**C**) 0–5 min and (**D**) 5–10 min after SSNA1 addition. (**E**) Cumulative distribution plots of SSNA1 single-molecule dwell times at 0–5 min and 5–10 min post-addition of SSNA1 for both the single-molecule control (blue dots) and spiking (purple dots) conditions.

The online version of this article includes the following video, source data, and figure supplement(s) for figure 4:

**Source data 1.** An Excel sheet containing numerical data for the quantification of single-molecule SSNA1 dwell times and SSNA1 fluorescence intensity, as presented in Figure 4 and Figure 4 - figure supplement 1.

**Figure supplement 1.** SSNA1 coats GMPCPP-stabilized microtubules.

**Figure 4—video 1.** SSNA1 coats GMPCPP-stabilized microtubules.

https://elifesciences.org/articles/67282/figures#fig4video1

activity (*Tan et al., 2019*; *Figure 5—figure supplement 2*). We assessed GFP-spastin's localization and activity on a mixed population of Taxol-stabilized microtubules that were either coated or not coated with 2 μM 647-SSNA1 (see Materials and methods for details). First, we determined whether SSNA1 prevented spastin binding by adding 25 nM GFP-spastin to the microtubules in the presence of 1 mM AMPPNP, a non-hydrolyzable ATP analog that allows spastin binding, but not severing of microtubules. We observed that GFP-spastin bound to both the SSNA1-coated and non-coated microtubules (*Figure 6A*). Quantitative fluorescence intensity analysis of the microtubules in the field of view confirmed that the GFP-spastin intensity was not significantly different between the populations of SSNA1-coated and non-coated microtubules (*Figure 6B, C*). Thus, SSNA1 does not prevent spastin from binding to microtubules under these conditions. We then assessed the effect of SSNA1 on GFP-spastin's microtubule severing activity by introducing 25 nM GFP-spastin to the microtubules in the presence of 1 mM ATP. Similar to our results with the unlabelled full-length spastin, we observed that the non-coated microtubules were severed within a few minutes, while the microtubules that were coated with SSNA1 were protected against spastin-induced severing (*Figure 6D–H*, *Video 6*). Spastin still localized to both SSNA1-coated and non-coated microtubules in the presence of ATP (*Figure 6D, E*). Taken together, we conclude that SSNA1 on microtubules inhibits spastin-mediated severing, despite permitting spastin's localization to the microtubule lattice.

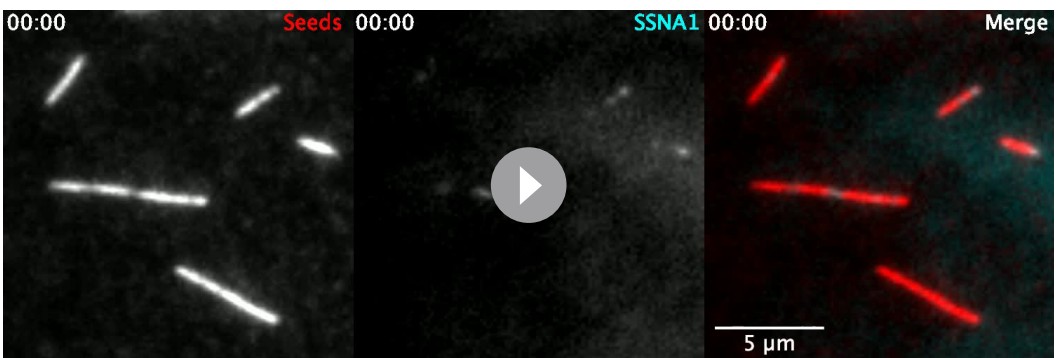

**Video 4.** SSNA1 detects spastin-induced microtubule lattice damage. GMPCPP-stabilized microtubules were pre-incubated with 100nM spastin to generate damage sites on the microtubule lattice and 5µM 647-SSNA1 was introduced at t=0min. Seeds are in red, SSNA1 is in cyan. Time is in min:s. Scale, 5µm. Playback, 20 fps.SSNA1 protects microtubules against spastin-mediated microtubule severing.
https://elifesciences.org/articles/67282/figures#video4

## Discussion

SSNA1 plays important roles in several fundamental, microtubule-based cellular processes including cilia formation, cell division, and axonal branching (*Basnet et al., 2018*; *Goyal et al., 2014*; *Lai et al., 2011*; *Pfannenschmid et al., 2003*; *Schoppmeier et al., 2005*). Despite an appreciation of the biological functions of SSNA1, an understanding of its direct effects on microtubules remained lacking. In this study, using in vitro reconstitution approaches, we explored the effects of SSNA1 on microtubules

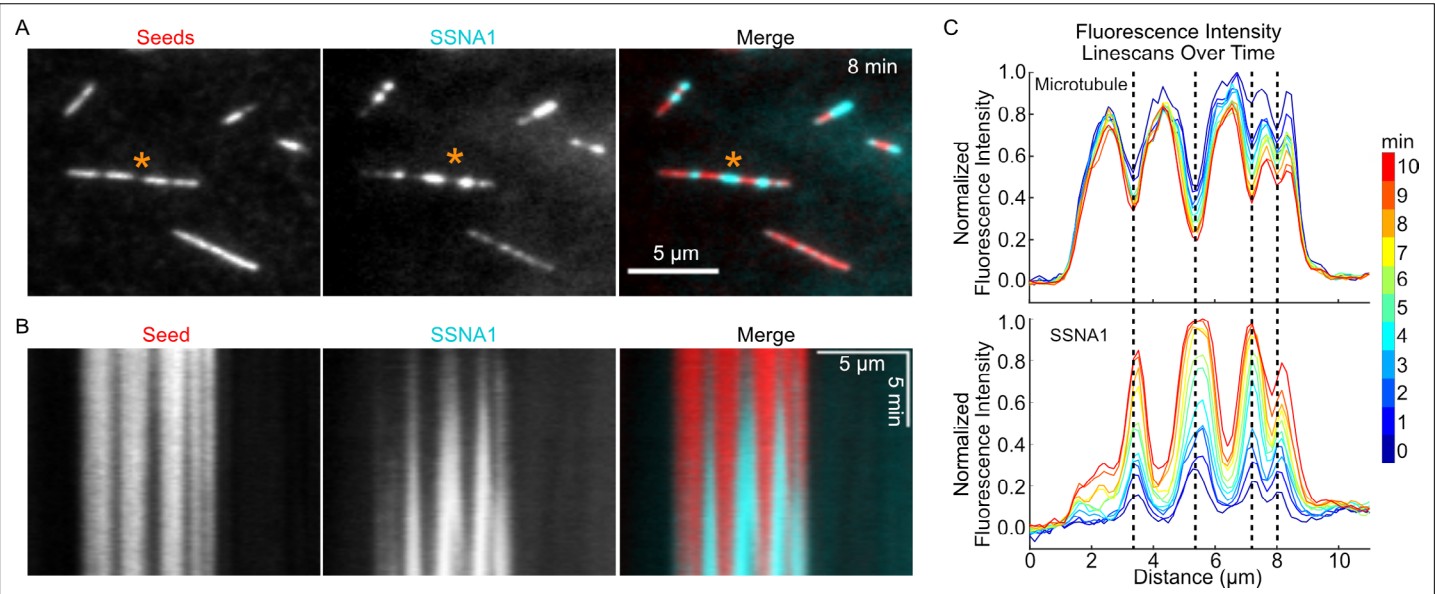

**Figure 5.** SSNA1 detects microtubule lattice damage. (**A**) Representative images of GMPCPP-stabilized microtubules (red) that were pre-incubated for 5 min with 100 nM spastin and 1 mM ATP and then subsequently incubated with 5 µM 647-SSNA1 (cyan). The images shown are from 8 min after SSNA1 addition. The orange asterisks indicate the microtubule used for kymograph and linescan analysis. (**B**) Kymographs showing SSNA1 localization to sites of spastin-induced microtubule damage. (**C**) Corresponding linescan analysis of the microtubule and SSNA1 intensities every minute from 0 min to 10 min after the introduction of SSNA1.

The online version of this article includes the following source data and figure supplement(s) for figure 5:

**Source data 1.** An Excel sheet containing numerical data for the quantification of the SSNA1 and seed fluorescence intensity linescans as presented in Figure 5 and Figure 5 - supplement 1.

**Figure supplement 1.** SSNA1 detects sites of microtubule lattice damage on GMPCPP-stabilized and Taxol-stabilized microtubules.

**Figure supplement 2.** Spastin protein purification.

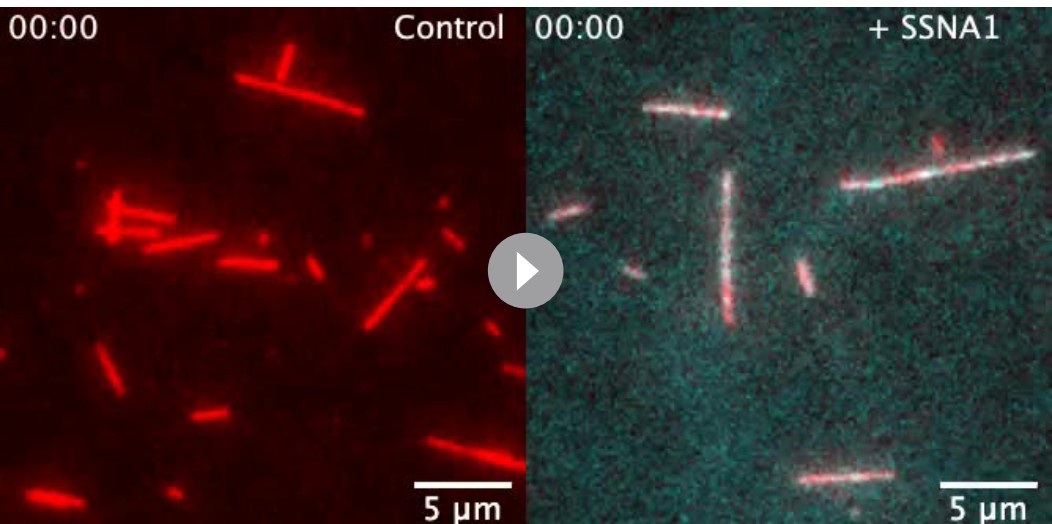

**Video 5.** SSNA1 protects microtubules against spastin-mediated microtubule severing. Control (left) and SSNA1-coated (right) microtubules were incubated with 200nM full-length human 6His-MBP-spastin. Microtubules were pre-incubated with either SSNA1 storage buffer (control) or 1µM 488-SSNA1 for 10min prior to the addition of spastin. Spastin was introduced at t=0min. Scale, 5µm.
https://elifesciences.org/articles/67282/figures#video5

and found that SSNA1 robustly stabilizes dynamic microtubules and detects sites of lattice damage, occurring both naturally and induced by spastin.

Microtubule-stabilizing proteins have critical functions in regulating microtubules in cells. One of the best studied classical stabilizing MAPs is tau, known for its roles in neurons and involvement in neurodegenerative diseases (*Barbier et al., 2019*; *Gao et al., 2018*; *Iqbal et al., 2016*; *Morris et al., 2011*). Studies in cells and in vitro report that tau forms oligomers on the outer microtubule surface (*Al-Bassam et al., 2002*), and stabilizes microtubules against depolymerization (*Drechsel et al., 1992*; *Prezel et al., 2018*; *Ramirez-Rios et al., 2016*). Similarly, SSNA1 forms fibrils that bind longitudinally on microtubules (*Basnet et al., 2018*; *Rodríguez-Rodríguez et al., 2011*). Our work demonstrates that SSNA1 simultaneously modulates all four parameters of microtubule dynamic instability—slowing down the rates of growth, shrinkage, and catastrophe, and promoting rescue. Thus, SSNA1 is a potent microtubule-stabilizing protein and this activity likely underlies its cellular function.

In contrast to tau, which has been reported to promote microtubule growth (*Drechsel et al., 1992*), the growth rate slows down in the presence of SSNA1. One way to slow down microtubule growth is through sequestration of soluble tubulin—this is the mechanism employed by Op18/Stathmin (*Arnal et al., 2000*; *Belmont and Mitchison, 1996*; *Cassimeris, 2002*; *Gupta et al., 2013*; *Steinmetz, 2007*). However, tubulin sequestration would affect the growth rate of all microtubules simultaneously; this is not the case with SSNA1, where the onset of slowdown occurs at different times for individual microtubules. Instead, we find that suppression of microtubule growth rate correlates with the progressive SSNA1 accumulation on microtubule ends over time. Microtubule growth slowdown may be a consequence of perturbations in the dynamic end structure, including incomplete tubules, exposed protofilaments or ragged microtubule ends. Indeed, a slowdown in growth is typically observed just prior to microtubule catastrophe (*Farmer et al., 2021*; *Maurer et al., 2014*). Notably, previous data from cryo-ET experiments indicate that SSNA1 can bind to partial tubule structures, as it supports the growth of protofilaments away from the mother microtubule (*Basnet et al., 2018*). We note that, in our study, we did not observe SSNA1-induced microtubule branching events previously reported in conditions where microtubules were grown in the presence of GMPCPP and much higher concentrations of SSNA1 from different species (*Basnet et al., 2018*). Nevertheless, both previous and our current studies raise the possibility that SSNA1 recognizes specific dynamic end structures that are incompatible with continued unperturbed growth.

It has recently been proposed that taxanes, microtubule-stabilizing compounds widely used in cancer therapy, recognize microtubule ends that are in a pre-catastrophe state (*Rai et al., 2020*).

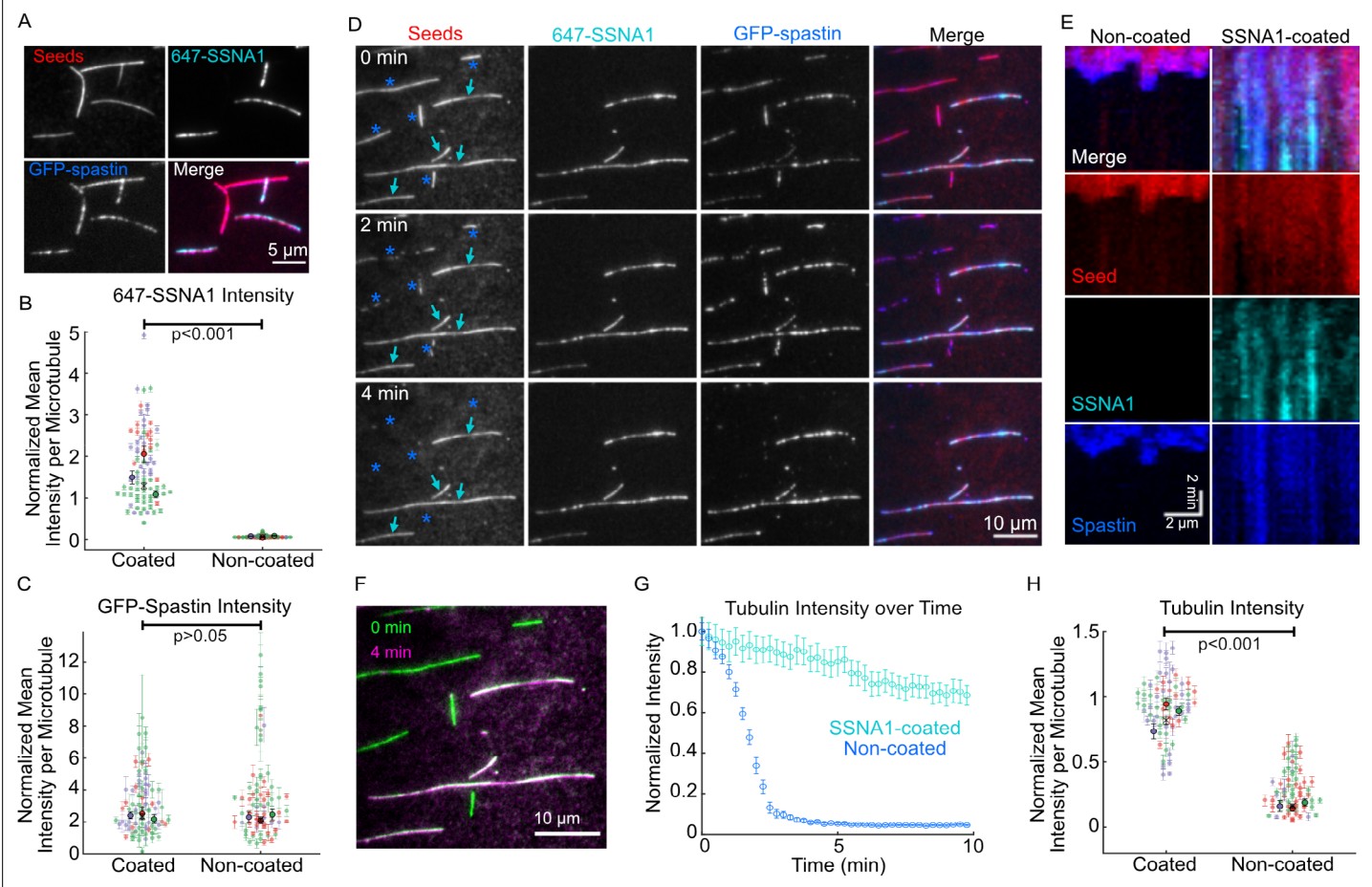

**Figure 6.** SSNA1 protects microtubules against severing by spastin. (**A**) Representative images of a mixed population of Taxol-stabilized microtubules that were either coated or not coated with 647-SSNA1 and treated with 25 nM GFP-spastin in the presence of 1 mM AMPPNP. (**B**) Quantification of the SSNA1 fluorescence intensities on SSNA1-coated (N=79) and non-coated (N=72) microtubules, p<0.001, unpaired t-test. (**C**) Quantification of the spastin fluorescence intensities on SSNA1-coated (N=79) and non-coated (N=72) microtubules, p=0.4, unpaired t-test. The colors in (**B**) and (**C**) represent different experimental repeats, the larger markers are mean ± SD for each experimental day. X marks the overall mean. (**D**) Representative images of a mixed population of microtubules (red) that were either coated or not coated with 2 μM 647-SSNA1 (cyan) prior to the introduction of 25 nM GFP-spastin (blue). The images are from 0 min, 2 min, and 4 min post-addition of spastin and 1 mM ATP. The cyan arrows indicate microtubules that were coated with SSNA1 and the blue asterisks were non-coated. (**E**) Representative kymographs of a non-coated (left) and SSNA1-coated (right) microtubules. (**F**) Overlay of the tubulin signal for the field of view in (**D**) at 0 min (green) versus 4 min (magenta), showing the loss of non-SSNA1-coated microtubules. (**G**) An example trace of tubulin fluorescence intensity over time for non-coated (blue) and SSNA1-coated (gray) microtubules in one field of view. Error bars represent SE. (**H**) Tubulin intensity for a time point where control (non-coated) microtubules reach an average of 20% of initial intensity. p<0.001, unpaired t-test. The colors represent different experimental repeats, the larger markers are mean ± SE for each experimental day. X marks the overall mean. N=66 microtubules per condition. Data are from three independent experimental repeats.

The online version of this article includes the following source data and figure supplement(s) for figure 6:

**Source data 1.** An Excel sheet containing numerical data for the quantification of the SSNA1, GFP-spastin and tubulin fluorescence intensities as presented in Figure 6.

**Figure supplement 1.** SSNA1 protects microtubules against full-length human spastin-mediated severing.

Significantly, the behavior of SSNA1 on dynamic microtubules is highly reminiscent of taxanes: at sub-saturating concentrations, taxanes accumulate at growing microtubule ends following growth perturbations, and form persistent patches that stabilize the microtubule lattice (*Rai et al., 2020*). While taxanes show a preference for GMPCPP-grown parts of the microtubule lattice, which mimic the extended GTP-tubulin conformation, we do not see enhanced localization of SSNA1 on the GMPCPP-lattice regions. Therefore, we do not think that nucleotide state recognition is the primary mechanism for SSNA1 localization. Electron microscopy data demonstrate that taxanes accumulate on incomplete microtubule lattice structures (*Rai et al., 2020*). Similarly, we find that stretches of

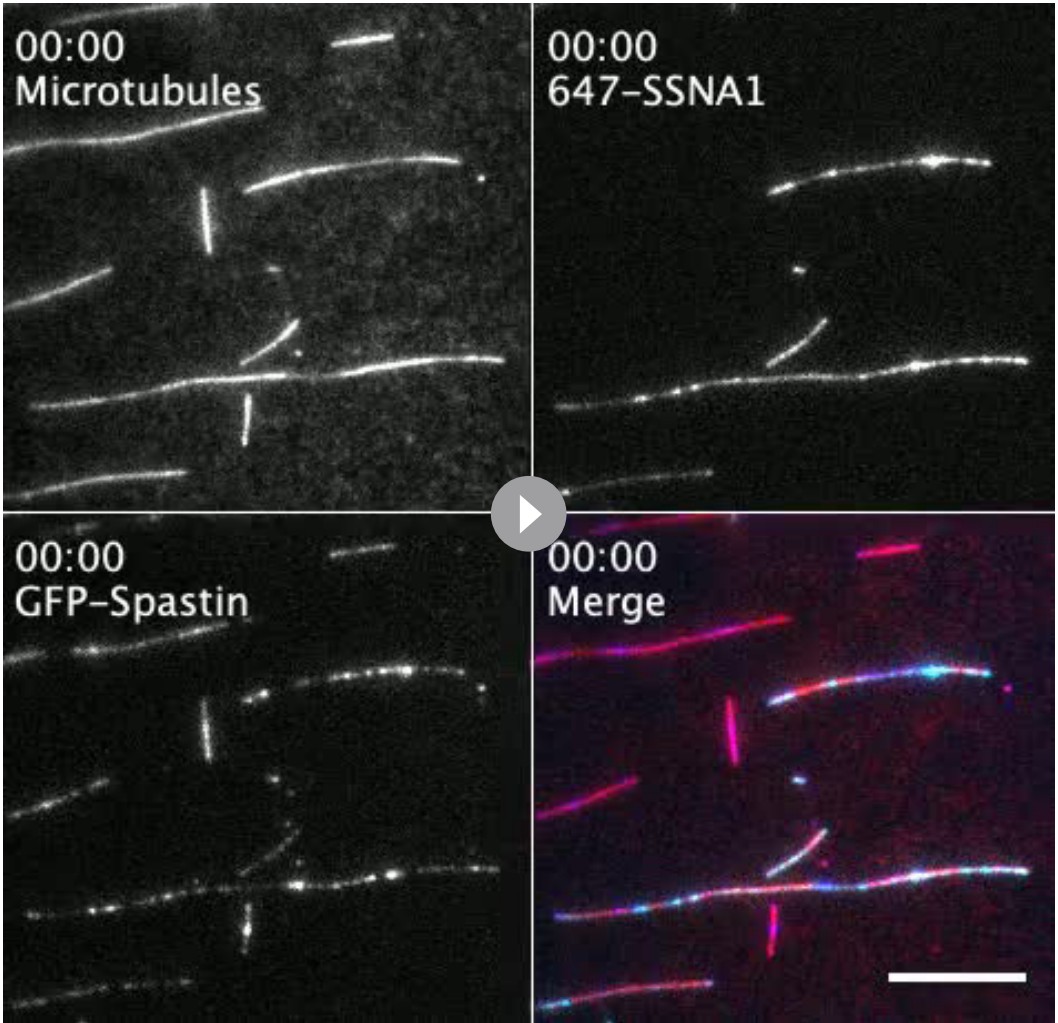

**Video 6.** SSNA1 prevents spastin severing but not binding. A mixed population of Taxol-stabilized microtubules (red) were either coated or not coated with 647-SSNA1 (cyan) and treated with 25 nM GFP-spastin (blue) in the presence of 1 mM ATP. Time is in min:s. Scale, 10 µm. Playback, 10 fps.
https://elifesciences.org/articles/67282/figures#video6

SSNA1 accumulate on dimmer regions of microtubule lattices (i.e., incomplete tubules) and growing microtubule ends. Thus, our data support a mechanism in which SSNA1 binding is determined by the structure of the microtubule lattice. Furthermore, we demonstrate that SSNA1 detects sites of spastin-mediated lattice damage. Hence, like taxanes, SSNA1 senses microtubule lattice damage.

Several lines of evidence support an important functional interplay between SSNA1 and spastin: SSNA1 is a binding partner of spastin (*Errico et al., 2004*); SSNA1 and spastin colocalize in dividing cells (*Goyal et al., 2014*); SSNA1 and spastin promote axonal branching in developing neurons (*Basnet et al., 2018*; *Goyal et al., 2014*; *Yu et al., 2008*); and SSNA1's putative spastin-binding domain is required to promote axonal branching (*Goyal et al., 2014*). Our data demonstrate that SSNA1 can both inhibit spastin activity and detect spastin-induced damage, further highlighting the interplay between the two proteins. Similarly, tau condensates on microtubule lattices were recently found to protect against severing by both spastin and katanin (*Siahaan et al., 2019*; *Tan et al., 2019*). SSNA1's microtubule-stabilizing activity is similar to that of CAMSAP2 and CAMSAP3: minus-end stabilizing proteins that associate with new minus ends and form stretches of stabilized lattice (*Jiang et al., 2014*). Interestingly, katanin interacts with CAMSAP2 and CAMSAP3 and limits the length of CAMSAP2-stretches on the microtubule minus ends. Thus, balancing the activity of microtubule severing and stabilizing proteins may represent a general mechanism for regulating microtubule number and mass in cells. Perhaps counterintuitively, spastin is implicated in microtubule network

amplification by generating new microtubules fragments after severing (*Kuo and Howard, 2021*; *Kuo et al., 2019a*; *Kuo et al., 2019b*; *Vemu et al., 2018*). Whether synergy between SSNA1's stabilizing activity and spastin's severing activity is critical for microtubule network amplification in cells presents an important area for future work.

Microtubule dynamics and network organization are influenced by the properties of both the microtubule end and lattice. Herein, we have identified SSNA1 as a microtubule-stabilizing protein and a sensor of microtubule damage. Our work thus provides mechanistic insight into an important but understudied microtubule regulatory protein with essential functions in cilia, cell division, and developing neurons.

## Materials and methods
### DNA constructs
The cDNA encoding human SSNA1 (NM_003731.2) with an N-terminal 6xHis-tag in a pReceiver-B01 vector was purchased from GeneCopoeia, Rockville, MD (product ID: Q0661). The cDNA encoding human spastin (NM_014946.3) was purchased from GeneCopoeia (product ID: U1177) and was subcloned into a modified pET vector containing N-terminal 6xHis and MBP tags; the pET MBP His6 LIC cloning vector (2Cc-T) was a gift from Scott Gradia (Addgene plasmid # 37237; http://n2t.net/addgene:37237; RRID: Addgene_37237). The cDNA encoding His-Strep-sfGFP-Spastin(del227) was a gift from the R.McKenney Laboratory (University of California – Davis) (*Tan et al., 2019*).

### Protein preparation
Bovine brain tubulin was purified using cycles of polymerization and depolymerization using the high-molarity PIPES method (*Castoldi and Popov, 2003*). Tubulin was labeled with tetramethylrhodamine (TAMRA), Alexa Fluor 488 and Alexa Fluor 647 dyes (Thermo Fisher Scientific, Waltham, MA) according to the standard protocols and as previously described (*Gell et al., 2010*; *Hyman et al., 1991*). Fluorescently labeled tubulin was used at a ratio between 5% and 10% of the total tubulin.

Human 6His-SSNA1 was expressed in BL21 DE3 Gold cells in Studier autoinduction media (Teknova, Hollister, CA; cat. #3S2000) for 96 hr. Expression cell pellets were lysed for 1 hr at 4°C in 50 mM HEPES (pH 7.5), 150 mM NaCl, 10% (v/v) glycerol, 10 mM imidazole, and 1 mM DTT and supplemented with 1 mg/ml lysozyme, 10 mg/ml PMSF, EDTA-free protease inhibitors (Roche, Basel, Switzerland), and 25 U/ml Pierce universal nuclease (Invitrogen). The crude lysate was sonicated on ice and clarified by centrifugation for 30 min at 4°C and 35,000 rpm in a Beckman L90K Optima and 50.2 Ti rotor (Beckman, Brea, CA). The clarified lysate was applied to a HisTrapHP column (Cytiva, Marlborough, MA) according to the manufacturer's protocol and eluted with 50 mM HEPES (pH 7.5), 150 mM NaCl, 10% (v/v) glycerol, 1 mM DTT and a linear gradient of 50–500 mM imidazole. The eluted protein was buffer exchanged using a PD-10 desalting column (Cytiva) into 20 mM HEPES (pH 7.5), 150 mM NaCl, 10% (v/v) glycerol, and 1 mM DTT. Purified SSNA1 was labeled using Alexa Fluor 488 and Alexa Fluor 647 Microscale Protein Labeling Kits (Thermo Fisher Scientific, cat. #A30006 and #A30009) according to the manufacturer's instructions. Protein purity was assessed by SDS-PAGE and mass spectrometry analysis.

Human 6His-MBP-spastin was expressed in Rosetta(DE3) cells and purified using a protocol adapted from *Kuo et al., 2019b*. Protein expression was induced with 0.5 mM IPTG and expressed overnight at 16°C. Cells were lysed for 1 hr at 4°C in 30 mM HEPES (pH 7.4), 300 mM NaCl, 10 mM imidazole, 5% glycerol, 2 mM DTT, 10 µM ATP, and 2 mM DTT and supplemented with 1 mg/ml lysozyme, 10 mg/ml PMSF, and EDTA-free protease inhibitors. The crude lysate was sonicated on ice and then clarified by centrifugation for 30 min at 4°C and 35,000 rpm in a Beckman L90K Optima and 50.2 Ti rotor. Clarified lysates were applied to a HisTrapHP column (Cytiva) according to the manufacturer's protocol. His-tagged protein was eluted with 30 mM HEPES (pH 7.4), 300 mM NaCl, 10 µM ATP, 5% (v/v) glycerol, and 2 mM DTT and linear gradient of 50–500 mM imidazole. For storage, the buffer was exchanged to 20 mM HEPES (pH 7.4), 0.15 M NaCl, 5% glycerol, 0.5 mM DTT, 10 µM ATP, and PMSF using PD-10 desalting columns (Cytiva).

Human His-Strep-sfGFP-Spastin(del227) was expressed in *Escherichia coli* and purified by Ni-column affinity chromatography followed by gel filtration on a Superdex 200 column by GenScript Protein Expression Services. Purified GFP-Spastin protein was stored in 50 mM Tris-HCl, 500 mM

NaCl, 200 mM L-Arginine, 10% Glycerol, and pH 8.0. For use in experiments, the concentrated stocks of GFP-spastin protein were diluted ~20-fold in BRB80 and re-frozen.

All proteins were snap frozen in liquid nitrogen and stored at –80°C.

## Imaging assay

All imaging was performed using a Nikon Eclipse Ti microscope with a 100×/1.49 n.a. TIRF objective (Nikon, Tokyo, Japan), Andor iXon Ultra EM-CCD (electron multiplying charge-coupled device) camera (Andor, Belfast, UK); 488-, 561-, and 640 nm solid-state lasers (Nikon Lu-NA); HS-625 high speed emission filter wheel (Finger Lakes Instrumentation, Lima, NY); and standard filter sets. An objective heater was used to maintain the sample at 35°C. Microscope chambers were constructed as previously described (*Gell et al., 2010*). Briefly, 22×22 mm$^2$ and 18×18 mm$^2$ silanized coverslips were separated by strips of Parafilm to create narrow channels for the exchange of solution. Images were acquired using NIS-Elements (Nikon) with exposure times of 50–100 ms and at the frame rates specified in the methods.

## Microtubule dynamics

For microtubule dynamics experiments, GMPCPP-stabilized microtubules seeds were prepared according to standard protocols (*Chen and Doxsey, 2012*; *Gell et al., 2010*). Dynamic microtubule extensions were polymerized from surface-immobilized GMPCPP-stabilized templates as described previously (*Gell et al., 2010*). Imaging buffer containing soluble tubulin ranging from 9 µM tubulin, 1 mM GTP, and proteins at the concentrations indicated in the text were introduced into the imaging chamber. The imaging buffer consisted of BRB80 supplemented with 40 mM glucose, 40 µg/ml glucose oxidase, 16 µg/ml catalase, 0.5 mg/ml casein, 50 mM KCl, 10 mM DTT, and 0.1% methylcellulose. Dynamic microtubules were grown with or without unlabeled SSNA1 and imaged for 30 min (0.2 fps).

Quantification of microtubule dynamics parameters was performed using kymographs generated in Fiji (*Schindelin et al., 2012*) as described previously (*Zanic, 2016*). Catastrophe frequency was calculated by dividing the number of catastrophes by the total time spent in the growth phase. Rescue was calculated by dividing the number of rescues observed by the total shrinkage length. The error for catastrophe frequency and rescue per shrinkage length are counting errors. Microtubule dynamicity was calculated by summing the total length of growth and shrinkage and dividing by the total observation time (*Toso et al., 1993*). The error for dynamicity was calculated as pixel size (160 nm) multiplied by $\sqrt{N}/T$, where N is the number of points marked on the kymographs.

For the analysis of microtubule growth rate over time, microtubule end positions were determined using KymographClear and KymographDirect using tubulin channel (*Mangeol et al., 2016*). A custom MATLAB (The MathWorks, Natick, MA) code was used to determine growth rate as a function of time. Briefly, a linear function was fit to position and time data points within a 2-min window to determine the mean velocity. The window was then shifted to the next 2-min interval and fitting procedure was repeated until the end of the trajectory. To focus on the characterization of microtubule growth over time, the segments with negative velocity at the beginning of a trajectory were eliminated (i.e., segment contains shrinkage phase), such that each trajectory starts with a positive velocity segment, while subsequent segments with negative velocity were kept. For each 2-min segment, position data were color-coded based on the determined velocity and plotted as a function of time. For further analysis of growth segments, segments determined to have velocities smaller than –2.5 nm/s were classified as shrinking segments, and were not included. Segments from multiple microtubule trajectories for a given time window were grouped and the median for each time window was calculated. Similarly, the velocity-versus-time data points were color-coded based on the segment velocity.

## Microtubule nucleation

The templated microtubule nucleation experiments were based on *Wieczorek et al., 2015*. GMPCPP-stabilized seeds were incubated with tubulin concentrations ranging from 0 µM to 10 µM in the presence and absence of 2.5 µM SSNA1 and imaged every 15 s for 15 min. Nucleation was measured as the fraction of individual GMPCPP seeds that were observed to grow at least one microtubule extension of >3 pixels in length (480 nm) within 15 min. Analysis was performed on maximum projection images from the 15-min time-lapse videos. Data across the range of the tubulin concentrations were

fitted to the sigmoidal equation y(x)=xs/(C+xs) in MATLAB, as previously published (*Wieczorek et al., 2015*), where C is the half-maximal concentration at which nucleation occurs and s is the steepness of the curve. The errors were calculated as $\sqrt{N}$/total number of seeds, where N is the number of nucleated seeds, except when 0 seeds nucleated, then the error was estimated as 1/total number of seeds.

## SSNA1 localization on taxol-stabilized microtubules

Taxol-stabilized microtubules were prepared as follows: microtubules were grown with 32 µM tetra-rhodamine-labeled tubulin (25% labeled) for 30 min at 35°C and stabilized with 10 µM Taxol (Tocris, Minneapolis, MN; Cat. # 1097) in BBR80 buffer. The taxol-stabilized microtubules were then spun in an airfuge for 5 min at 20 psi and sheared with an 18-gauge needle. Taxol microtubules were bound to the coverslip surface of a flow cell using anti-Rhodamine antibody, then 5 µM 488-SSNA1 was introduced and the microtubules were imaged every 30 s for 60 min.

## SSNA1 localization on GMPCPP-stabilized microtubules

GMPCPP-stabilized microtubules adhered to coverslips were incubated with 2 µM Alexa-488-SSNA1 and imaged for 10 min (0.2 fps). The total SSNA1 fluorescence intensity along the total length of microtubule lattice in the microscope field of view excluding the background was measured in every frame (5 s interval) of the 10-min movie using Fiji. The background from the SSNA1 channel was excluded by creating a mask around the microtubules, applying the mask to the SSNA1 channel, and including only the areas occupied by microtubules in the SSNA1 intensity measurements.

## SSNA1 localization on dynamic microtubules

For SSNA1 localization experiments on dynamic microtubules, microtubules were grown from GMPCPP-microtubule seeds with 9 µM tubulin and 5 µM 488-SSNA1 and imaged for 60 min (0.2 fps).

For the mixed-lattice wash-in experiments (*Figure 2D*), dynamic microtubules were pre-grown from GMPCPP-microtubule seeds with 15 µM tubulin for 15 min and then the reaction was exchanged for 15 µM tubulin and 2.5 µM Alexa-488-SSNA1 (15% labeled). The microtubules were imaged for 60 min in total (2 min prior to exchange and 58 min after exchange) at 0.2 fps.

To quantify SSNA1 localization on growing microtubules over time, the microtubule end velocity was determined for 2-min segments, and the mean SSNA1 intensity at the microtubule end was determined using a custom MATLAB code. For each time frame (i.e., horizontal line on a kymograph) within 2-min segment: (i) mean solution background intensity was calculated within a 5-pixel-long region located three pixels away from the tracked end position, (ii) mean lattice intensity was calculated within a 5-pixel-long region on the microtubule lattice ending at the tracked end position, (iii) the mean solution background intensity was subtracted from mean lattice intensity to obtain mean SSNA1 intensity. Time frames that had <5 pixels available for background intensity calculation were eliminated. Next, an average SSNA1 intensity within 2 minute segment was calculated using single-time-frame (frame interval=5 s) mean SSNA1 intensities. The average SSNA1 intensity data points were color-coded with 2-min segment velocity and plotted as a function of time and velocity.

To determine the direction of SSNA1 stretch expansion at growing microtubule ends, the position of initial stretch initiation was identified by manual kymograph analysis. A 20-pixel (3.2 µm) SSNA1 intensity linescan centered on the initiation location along the microtubule was measured once a stretch reached a minimum length of 2 µm (6–14 min after initiation). SSNA1 intensity along the linescan was plotted relative to the intensity at the center point.

## Microtubule curvature analysis

To determine curvature at the microtubule ends, we first traced curled extensions from a maximum projection image of microtubules copolymerized with SSNA1 (N=17) using ImageJ plugin 'Kappa – Curvature Analysis.' The coordinates of traces and curvature data for each point were exported. Using MATLAB, we grouped the curvature data from all microtubules using 0.1 µm$^{-1}$ bin width. We then imported corresponding SSNA1 channel into MATLAB. We performed outlier analysis based on SSNA1 intensity for each pixel within a given curvature bin using MATLAB function 'isoutlier.' We then plotted intensity as a function of curvature with means and standard deviations from each curvature bin. Finally, we fitted a linear function to the mean curvature and mean intensity. This analysis was

repeated for curled taxol-stabilized microtubules using sum projection images from four fields of view (N=59).

### Single-molecule dwell time analysis

GMPCPP-seeds were incubated with either 5 nM 647-SSNA1 (100% labeled) alone (control) or 5 nM 647-SSNA and 1 µM 488-SSNA1 (spiking) and imaged for 5 min at 5 fps using maximum laser power and 50 ms exposure. The durations of the binding events were manually measured from kymographs and plotted in histograms and cumulative distribution function plots in MATLAB. Data were obtained for two consecutive 5-min segments (0–5 min and 5–10 min) after the introduction of SSNA1, moving to different fields of view to minimize photobleaching. The mean dwell times reported are arithmetic means ± SE. The association rates were calculated as the number of binding events per second per total length of the microtubule seeds in µm per nM.

### Microtubule severing and damage assays

For all experiments using spastin, the imaging buffer was supplemented with 1 mM ATP and 1 mM MgCl2 was included whenever spastin was used. For the microtubule severing assay, GMPCPP-stabilized microtubules were incubated with 1 µM 488-SSNA1 or SSNA1 storage buffer (control) for 10 min, then 200 nM spastin and 1 µM 488-SSNA1 was introduced into the flow cell and the microtubules were imaged for up to 30 min at 0.5 fps.

For the damage recognition assay, GMPCPP-stabilized microtubules were first incubated for 5 min with 100 nM spastin to generate microtubule lattice damage. The reaction was washed out with BRB80 and imaging buffer and then the damaged microtubules were incubated with 5 µM 647-SSNA1 and imaged every 10 s for 30 min.

### SSNA1-spastin interplay with mixed microtubule population

Taxol-stabilized, TAMRA-labeled microtubules were first attached to coverslip. 2 µM SSNA1 was added and incubated for 10 min. At the end of SSNA1 incubation, we added more microtubules into the flow chamber and allowed them to immobilize on coverslip surface. This resulted in two populations of microtubules that were either coated or non-coated with SSNA1 in the same field of view. To test whether SSNA1 prevents spastin binding, we added 25 nM GFP-spastin(del227) with 1 mM AMPPNP and imaged for 10 min every 15 s. To test whether SSNA1 prevents spastin-mediated severing, we then added 25 nM GFP-spastin(del227) with 1 mM ATP and imaged for 10 min every 15 s. The analysis was performed using ImageJ and MATLAB. First, we registered the image time series using ImageJ plugins 'Template Matching,' 'Image Stabilizer' and 'Image Stabilizer Log Applier' Next, we used ImageJ plugin 'JFilament' to trace microtubules using the first frame of the time series and exported the pixel coordinates. Background subtraction was then performed using a rolling ball with a 5-pixel radius in ImageJ. A binary mask was generated in MATLAB using the exported coordinates. The average intensity of SSNA1 and spastin per microtubule in the AMPPNP condition was averaged over time (between 3 min and 5 min, normalized with intensity values between 0 min and 1 min). Outlier analysis was then performed using MATLAB 'isoutlier' function, which identified four outliers each in populations of 83 and 76 microtubules, respectively. Finally, we calculated weighted mean and standard error for a population of microtubules using inverse square of standard error of intensity per microtubule. For the ATP condition, we first determined the time point at which normalized tubulin intensity fell below 20% for two consecutive frames for each field of view for the control condition (i.e., non-SSNA1 coated), and then determined the intensity of SSNA1-coated microtubules at the same time point. Outlier analysis was performed using MATLAB 'isoutlier' function as above, which identified one outlier in the population of SSNA1-coated microtubules out of 67 microtubules analyzed. Finally, we calculated weighted mean and standard error for a population of microtubules using inverse square of standard error of intensity per microtubule.

## Acknowledgements

The authors thank S Hall for help with protein purification, C Strothman for the MATLAB dynamics analysis script, and R McKenney (University of California Davis) for the GFP-spastin DNA construct. The authors thank H McDonald and the Vanderbilt Mass Spectrometry Research Center (MSRC) Cores for the mass spectrometry analysis, which was supported in part by Vanderbilt Ingram Cancer

Center Resource Share Scholarship 2020-2909607. The authors thank members of the Zanic lab and the Vanderbilt Microtubules and Motors Club for discussions and feedback. EJL acknowledges the support of the National Institutes of Health IBSTO training grant T32CA119925. MZ acknowledges support from the National Institutes of Health grant R35GM119552, and the National Science Foundation grant MCB2018661.

## Additional information

### Funding

| Funder | Grant reference number | Author |
| --- | --- | --- |
| National Institutes of Health | T32CA119925 | Elizabeth J Lawrence |
| National Institutes of Health | R35GM119552 | Marija Zanic |
| National Science Foundation | MCB2018661 | Marija Zanic |

The funders had no role in study design, data collection and interpretation, or the decision to submit the work for publication.

### Author contributions

Elizabeth J Lawrence, Conceptualization, Formal analysis, Investigation, Methodology, Writing - original draft, Writing – review and editing; Goker Arpag, Formal analysis, Investigation, Writing – review and editing; Cayetana Arnaiz, Investigation; Marija Zanic, Conceptualization, Funding acquisition, Supervision, Writing – review and editing

### Author ORCIDs

Elizabeth J Lawrence http://orcid.org/0000-0001-9543-8678
Goker Arpag http://orcid.org/0000-0002-6893-2678
Cayetana Arnaiz http://orcid.org/0000-0001-5069-1469
Marija Zanic http://orcid.org/0000-0002-5127-5819

### Decision letter and Author response

Decision letter https://doi.org/10.7554/eLife.67282.sa1
Author response https://doi.org/10.7554/eLife.67282.sa2

## Additional files

### Supplementary files
• Transparent reporting form

### Data availability
All data generated or analyzed during this study are included in the manuscript and supporting files.

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
