## [Editor Report]

In this manuscript, Lawrence et al. investigate the direct effects of the microtubule-associated protein, Sjögren's Syndrome Nuclear Autoantigen 1 (SSNA1), on microtubule dynamics and damage using purified proteins and TIRF microscopy. The authors show that SSNA1 is a microtubule stabilizing protein that acts to slow rates of growth and shrinkage, and promote rescue. Furthermore, SSNA1 serves as a sensor of microtubule damage and protects microtubules from the microtubule severing enzyme, spastin. This paper will be of broad interest to scientists interested in cytoskeletal cell biology.

---

## [Decision Letter]

**Decision letter after peer review:**

Thank you for submitting your article "SSNA1 stabilizes dynamic microtubules and detects microtubule damage" for consideration by *eLife*. Your article has been reviewed by 3 peer reviewers, one of whom is a member of our Board of Reviewing Editors, and the evaluation has been overseen by Suzanne Pfeffer as the Senior Editor. The following individual involved in review of your submission has agreed to reveal their identity: Marcus Braun (Reviewer #3).

Essential revisions:

1. Please provide a saturation curve for SSNA1 on GMPCPP and taxol stabilized microtubules to inform the reader of SSNA1 binding affinities. Since the SSNA1 concentrations used are very high, it would be good to show that, under the conditions used, SSNA1 is binding reversibly to the microtubule lattice (i.e. doesn't come in protein clusters sticking non-specifically). Also, provide quantification of certain parameters (such as catastrophe frequency) of the microtubule dynamics experiments at physiological concentrations <100 nM. The conclusion might be that catastrophe and shrinking are affected, while growth and rescue are not, making the ones shown to be more convincing/relevant.

2. The question of SSNA1 localization and non-uniform-distribution need to be addressed, and, importantly, quantifications need to be provided. Please provide quantification for Figures 2C, 2D, and S4. Please see comments from Reviewer 1 and 3, as they had similar comments for 2D.

3. All three reviewers were curious about whether you observed branched nucleation in any of your assays. Please provide data and/or discussion for your observations in more detail in the paper. Please refer to the additional comments on this subject from all three reviewers.

4. All three reviewers asked if SSNA1 binds the microtubule cooperatively. This could be tested by performing titration experiments or by comparing the dwell times of SSNA1 in regions of low density to the dwell times in regions of high density (FRAP or single molecule fluorescent imaging in a background of non-labeled SSNA1). Either experiment could also provide more information as to accumulation at specific sites on the microtubule.

5. For Figure 3, please provide images of the SSNA1 for these assays.

6. There are a few concerns that should be addressed either textually or with additional data with regards to the proteins used in the study: 1) Why is SSNA1 not expressed with a GFP fusion as published for many other proteins from the same lab? Does it interfere with function or solubility? 2) Please provide gel filtration chromatograms for SSNA1 and spastin, as well as gels for the purified spastin and stathmin proteins.

7. All three reviewers agree that the spastin/damage experiments could be the most exciting conceptual advance of this paper, but at the moment this part of the paper is underdeveloped. The major requests are: Reviewer 1: does SSNA1 prevent spastin binding or severing? The authors should observe spastin in the absence of ATP, both in the absence or presence of SSNA1 to determine if spastin is prevented from binding the MT in the presence of SSNA1. Also, the authors should include images of the spastin on the MT for Figure 4A. There are some sites that appear to become severed even in the presence of SSNA1 and it would be helpful to the reader to see where the spastin is accumulating (or not accumulating). The authors should show a timelapse of the entire process. For example, they could take images every ~2 min of spastin + MTs, showing both channels, then wash out spastin and flow in SSNA1 and take images again every ~1-2 min to watch the accumulation of SSNA1 at the sites of damage. (This reviewer also asks to look for tubulin incorporation at these damage sites, but this is not necessary). Reviewer 2: Can you exclude that SSNA1 binds to spastin at the damage sites and not to damaged microtubules directly? If you introduce both SSNA1 and spastin at the same time, is this as protective? Does SSNA1 oligomerise to achieve protective function?

---

## [Author Response]

Essential revisions:1. Please provide a saturation curve for SSNA1 on GMPCPP and taxol stabilized microtubules to inform the reader of SSNA1 binding affinities. Since the SSNA1 concentrations used are very high, it would be good to show that, under the conditions used, SSNA1 is binding reversibly to the microtubule lattice (i.e. doesn't come in protein clusters sticking non-specifically). Also, provide quantification of certain parameters (such as catastrophe frequency) of the microtubule dynamics experiments at physiological concentrations <100 nM. The conclusion might be that catastrophe and shrinking are affected, while growth and rescue are not, making the ones shown to be more convincing/relevant.

We have included saturation curves for SSNA1 binding to stabilized microtubules in Figure 4 – Supplemental Figure 1. We find that the build-up of SSNA1 on microtubules is linear and gradual and does not occur by the landing of large protein clusters or aggregates (Figure 4 – Supplemental Figure 1). Furthermore, we have significantly expanded our single molecule analysis to determine SSNA1 landing rates and dwell times, both in single-molecule, as well as in spiking conditions, as suggested by the reviewers. Our data show that single molecules of SSNA1 bind reversibly to the GMPCPP-stabilized microtubule lattices (Figure 4A, B). In the spiking conditions, where single molecules of 647-SSNA1 (5 nM) are visualized binding to microtubules in the background of 1 µM 488-SSNA1 (Figure 4C, D), we find that SSNA1 still binds in a reversible manner as single molecules and not as large protein aggregates. Interestingly, we find that the dwell time of SSNA1 molecules is increased in the spiking condition compared to the true single molecule condition, suggesting interactions between SSNA1 molecules in the higher concentration conditions. Moreover, both the dwell times and the landing rates of SSNA1 in spiking conditions increase over time (Figure 4E). Thus, our results demonstrate that SSNA1 binding to microtubules is cooperative. This finding is consistent with previous reports that SSNA1 forms fibrils which can localize along the microtubule lattice (Rodriguez-Rodriguez et al. 2011, Basnet et al. 2018).

Furthermore, we appreciate the concern about the physiologically-relevant SSNA1 concentrations. To that end, please note that the average cellular concentration of SSNA1 has been reported to be: ~200 nM in Hela cells (Itzhak et al. 2016) (http://www.mapofthecell.org/), which is within the range of conditions we investigated in our titration experiments (125 nM to 3 µM). At these lower concentrations of SSNA1, we indeed find that the major effect on microtubule dynamicity is due to SSNA1’s effects on microtubule catastrophe and shrinkage rate (Figure 1EF), rather than growth and rescue (Figure 1 – Supplemental Figure 2). Nevertheless, imaging of SSNA1 in HeLa cells demonstrated that SSNA1 localizes to the centrosomes in these cells (Goyal et al. 2014). Furthermore, in neurons SSNA1 localization is restricted to branch points (Goyal et al. 2014, Basnet et al. 2018), while in cells with primary cilia, SSNA1 is localized to the basal bodies (Lai et al. 2011). Hence, we argue that the local concentrations of SSNA1 in physiologically-relevant conditions are likely to be much higher than 200 nM. As a back-of-the-envelope calculation, assuming a typical centrosome radius of 2 µm (see e.g. Baumgart et al. JCB 2019), the estimated centrosome volume is ~30 µm^3^, which is ~2 orders of magnitude smaller than the typical volume of the HeLa cell (~ 2600 µm^3^, https://bionumbers.hms.harvard.edu). Therefore, we think that understanding SSNA1 activity at higher effective concentrations is of physiological relevance, given its restricted subcellular localization. In particular, SSNA1 could help to nucleate and then stabilize the nascent microtubules in the centrosome, based on our findings that SSNA1 promotes nucleation and suppresses catastrophe at concentrations in the hundreds of nanomolar. Along those lines, please note that a previous in vitro study of SSNA1’s effects (Basnet et al. 2018) used significantly higher protein concentrations (30 µM SSNA1) and/or high concentrations of crowding agents (i.e. 200 nM SSNA1 with 7.5% PEG).

Nevertheless, to showcase the effects of SSNA1 at lower concentrations, as requested by the reviewers, we now present effects on catastrophe and shrinkage rates for concentrations below 1 µM SSNA1 in the revised main Figure 1, and have included the full titration results with all of the individual parameters of microtubule dynamics in the presence of up to 3 µM SSNA1 in the supplement (Figure 1 – Supplemental Figure 2). We have also added a representative kymograph showing the effects of 500 nM SSNA1 on microtubule dynamics in the main figure (Figure 1C, middle kymograph). Finally, we included a sentence on physiologically-relevant SSNA1 concentrations in the main text (lines 66-69).

2. The question of SSNA1 localization and non-uniform-distribution need to be addressed, and, importantly, quantifications need to be provided. Please provide quantification for Figures 2C, 2D, and S4. Please see comments from Reviewer 1 and 3, as they had similar comments for 2D.

We agree that quantification of these data is important to substantiate our findings. As such, we have now quantified SSNA1 intensity as a function of microtubule end curvature for both Taxolstabilized microtubules (Figure 3J, K, L, previously S4) and curved dynamic microtubule ends (Figure 3G, H, I, previously 2C). On Taxol-stabilized microtubules, which are naturally curved, our quantification demonstrates that SSNA1 localization is not enhanced on curved microtubule regions (Figure 3L). Furthermore, we find that, while SSNA1 is enriched on the curled growing microtubule ends, the SSNA1 intensity does not correlate with the degree of microtubule end curvature (Figure 3I). Taken together, our data are consistent with the model in which SSNA1 recognizes open microtubule lattice at growing microtubule ends, incorporates along the newly grown microtubule lattice and directly induces microtubule end curvature, but is not itself a sensor of microtubule curvature.

In addition, we have expanded our quantitative analysis of SSNA1 localization on dynamic microtubules. Our quantification of SSNA1 localization in dynamic wash-in experiments (Figure 3A-D, previously 2D) shows that SSNA1 does not preferentially localize to the stabilized GMPCPP-microtubule seeds (which would be expected if it recognized GTP-nucleotide state of tubulin), over dynamically-grown (GDP-tubulin) extensions. Our analysis shows that stretches of SSNA1 accumulation start at the growing microtubule end and expand in the direction of microtubule growth (Figure 3F), preferentially decorating the portions of microtubule lattice grown in the presence of SSNA1 (Figure 3B, D). Our expanded quantitative analysis of SSNA1 localization is now presented in a new figure of the revised manuscript (Figure 3).

3. All three reviewers were curious about whether you observed branched nucleation in any of your assays. Please provide data and/or discussion for your observations in more detail in the paper. Please refer to the additional comments on this subject from all three reviewers.

Basnet et al., 2018 previously observed microtubule branching when microtubules were grown in the presence of GMPCPP and very high concentrations of SSNA1. Interestingly, in none of our experiments, under our particular experimental conditions, did we observe any microtubule branching events. Possible reasons for this include our use of human SSNA1, as opposed to the *Chlamydomonas reinhardtii* SSNA1 used in Basnet et al., or that the microtubules were grown with GMPCPP in Basnet et al., while we used GTP. Furthermore, Basnet et al., used much higher concentrations of SSNA1 (30 µM – 75 µM) than used in our manuscript (100 nM – 5 µM, which could explain the additional branching activity observed. We have now added a sentence to this effect in the discussion:

“We note that, in our study, we did not observe SSNA1-induced microtubule branching events previously reported in conditions where microtubules were grown in the presence of GMPCPP and much higher concentrations of SSNA1 from different species (Basnet et al. 2018).” (lines 235-238)

4. All three reviewers asked if SSNA1 binds the microtubule cooperatively. This could be tested by performing titration experiments or by comparing the dwell times of SSNA1 in regions of low density to the dwell times in regions of high density (FRAP or single molecule fluorescent imaging in a background of non-labeled SSNA1). Either experiment could also provide more information as to accumulation at specific sites on the microtubule.

Prompted by reviewers’ comments, we have now significantly expanded our single-molecule analysis to include ‘spiking’ experiments, where we analyzed the dwell times and landing rates of single SSNA1 molecules, both alone and in the background of 1 µM SSNA1. From these experiments, we found that the dwell times of SSNA1 were extended in the presence of excess SSNA1 (from 10 s ± 2 s, SE, N=107 to 39 s ± 4 s, SE, N=294, measured in the 2^nd^ 5 minutes after SSNA1 introduction, p<0.001, Wilcoxon rank test). Additionally, we found that both the dwell times and the landing rates increased over time. Thus, we can indeed conclude that SSNA1 binding to microtubules is cooperative. These data are now presented in a new figure in the revised manuscript (Figure 4).

Furthermore, our expanded analysis of regions displaying enhanced SSNA1 localization (new Figure 3) showed that stretches of SSNA1 initiate at growing microtubule ends (either plus or minus), increase over time and always extend only in the direction of microtubule growth (either plus or minus). SSNA1 stretches appear on many, but not all, growing microtubules, and the initial binding events on different microtubule ends occur at distinct times. Notably, SSNA1 patches can be resolved, such that a microtubule continues with dynamic growth beyond the extension of the SSNA1 patch, indicating a potential damage-repair event. Thus, rather than the site of initial SSNA1 localization serving as a seed for subsequent SSNA1 fibril elongation in either direction, our results suggest that SSNA1 binding actively detects microtubule damage as it is built into the microtubule lattice. Taken together, we can conclude that SSNA1 indeed detects microtubule damage occurring at growing microtubule ends, and that at higher concentrations SSNA1 molecules interact with each other, prolonging the dwell time of individual molecules, potentially through formation of SSNA1 fibrils, as previously reported (Basnet et al., 2018).

Our new extended analysis of SSNA1 single-molecule and stretch localization is presented in newly added Figures 3 and 4 and the corresponding discussion in the main text of the manuscript (lines 103-159).

5. For Figure 3, please provide images of the SSNA1 for these assays.

The experiments in former Figure 3 (catastrophe induced by tubulin dilution and stathmin) were performed with unlabeled SSNA1 and therefore SSNA1 was not imaged. In restructuring the manuscript from its initial focus on microtubule dynamics towards the SSNA1’s microtubule damage recognition, as suggested by the reviewers, we have decided not to pursue additional experiments focusing on the anti-catastrophe effects. We find that this previous result does not provide additional insights relevant to the current manuscript, and have thus omitted it in the revised version. Nevertheless, our expansion of quantitative SSNA1 localization analysis (new Figure 3) now provides additional new insights into SSNA1 localization on microtubules.

6. There are a few concerns that should be addressed either textually or with additional data with regards to the proteins used in the study: 1) Why is SSNA1 not expressed with a GFP fusion as published for many other proteins from the same lab? Does it interfere with function or solubility? 2) Please provide gel filtration chromatograms for SSNA1 and spastin, as well as gels for the purified spastin and stathmin proteins.

SSNA1 is approximately 15 kDa and forms fibrils by a head-to-tail interaction between SSNA1 dimers (Basnet et al., 2018). Since GFP is approximately 27 kDa (almost twice as large as SSNA1), we reasoned that it could interfere with SSNA1 protein function. Furthermore, GFP tags are usually expressed on the C- or N-terminus of the protein; we feared this may impede the ability of SSNA1 to self-oligomerize into fibrils since fibril formation depends on key residues in the N- and C- termini of SSNA1 (Basnet et al., 2018). For those reasons, we opted to chemically label SSNA1 with small Alexa-647 and Alexa-488 dyes. We have confirmed that this labeling procedure did not prevent SSNA1 fibril formation (Figure 2—figure supplement 2).

The protocols for purification of SSNA1 and spastin did not include a gel-filtration step (see Methods). Our protocols were based on the published protocols for SSNA1 and spastin purifications which did not include gel-filtration purification steps (Basnet et al., 2018, Tan et al. 2021). We note that our added GFP-spastin(del227) purification was performed by GenScript, and did include a gel-filtration step. We have now provided a gel showing the purified spastin and GFP-spastin proteins (Figure 5 – Supplemental Figure 2), in addition to the gel, Western Blot and mass-spec analysis of SSNA1 (Figure 1 – Supplemental Figure 1). Please note that stathmin is no longer used in the revised version of the manuscript.

7. All three reviewers agree that the spastin/damage experiments could be the most exciting conceptual advance of this paper, but at the moment this part of the paper is underdeveloped. The major requests are: Reviewer 1: does SSNA1 prevent spastin binding or severing? The authors should observe spastin in the absence of ATP, both in the absence or presence of SSNA1 to determine if spastin is prevented from binding the MT in the presence of SSNA1. Also, the authors should include images of the spastin on the MT for Figure 4A. There are some sites that appear to become severed even in the presence of SSNA1 and it would be helpful to the reader to see where the spastin is accumulating (or not accumulating). The authors should show a timelapse of the entire process. For example, they could take images every ~2 min of spastin + MTs, showing both channels, then wash out spastin and flow in SSNA1 and take images again every ~1-2 min to watch the accumulation of SSNA1 at the sites of damage. (This reviewer also asks to look for tubulin incorporation at these damage sites, but this is not necessary). Reviewer 2: Can you exclude that SSNA1 binds to spastin at the damage sites and not to damaged microtubules directly? If you introduce both SSNA1 and spastin at the same time, is this as protective? Does SSNA1 oligomerise to achieve protective function?

We agree with the reviewers that the spastin/SSNA1 experiments are of high interest and should be explored further. To this end, we have expressed and purified recombinant GFP-spastin, which allowed us to visualize spastin localization on microtubules that were either coated or noncoated with SSNA1 under severing and non-severing conditions. The mixed microtubule population was achieved by flowing microtubules into the imaging chamber and incubating with SSNA1 and then flowing in a second batch of microtubules which were not coated with SSNA1.

First, we asked whether SSNA1 prevents spastin binding to microtubules by introducing spastinGFP with 1 mM AMPPNP to the mixed microtubule population. We found that SSNA1 did not prevent spastin-GFP binding to microtubules. Next, we asked whether SSNA1 inhibits spastinmicrotubule binding under severing conditions. When GFP-spastin was introduced to the mixed microtubule population in the presence of 1 mM ATP, we found that, while GFP-spastin localized to all microtubules, only those that were not coated with SSNA1 were severed. Therefore, from these new experiments with labelled spastin, we now conclude that SSNA1 inhibits spastin severing but does not prevent spastin binding to microtubules. These results are now presented in a new figure of the revised manuscript (Figure 6).

Our data observing SSNA1 localization on Taxol-stabilized microtubules demonstrates that SSNA1 recognizes spontaneously occurring damage sites on the microtubule lattice, as evidenced by enhanced SSNA1 localization at sites of lower tubulin signal (i.e. damage). Since these damage sites were generated without spastin, we are confident in concluding that SSNA1 specifically detects microtubule damage (rather than binding to spastin at damage sites). To emphasize this distinction, we now have a dedicated supplemental figure showing SSNA1 localization at damage sites on Taxol-microtubules, which also includes examples of SSNA1 enrichment at sites of spontaneous damage on GMPCPP-stabilized microtubules, and on dynamic GDP microtubule extensions (Figure 5-Supplemental Figure 1).

We agree with the reviewer that SSNA1 oligomerization could be involved in SSNA1’s protective activity. Indeed, our spiking analysis revealed that SSNA1 binding to microtubules is cooperative, so it is possible that formation of higher-order SSNA1 structures protects microtubules against severing. We have also performed a limited number of experiments in which SSNA1 and spastin were simultaneously introduced to microtubules (data not shown). Under the specific conditions tested (up to 200 nM spastin with 1 µM SSNA1), SSNA1 was not able to protect against spastin activity, which could further indicate that SSNA1 must first oligomerize to efficiently protect against spastin, or that the microtubule needs to be fully coated with SSNA1 in order to be protected (i.e. if spastin can access uncoated tubulin dimers in the lattice then it will remove them). Indeed, even in conditions with as low as 250 nM SSNA1, we could observe protection against spastin, as long as a preincubation step was included. Given the low association rates of SSNA1 to microtubules, this observation is consistent with the idea that higher-order structures or full microtubule coverage are needed. A possible way to test the hypothesis that SSNA1 fibril formation is necessary for microtubule protection would be to first polymerize SSNA1 fibrils and then introduce these onto microtubules along with spastin to assess severing activity, compared to when spastin is introduced with non-filamentous SSNA1. These technically challenging experiments will be the focus of subsequent research efforts in our lab.